# EGFR/ARF6 regulation of Hh signalling stimulates oncogenic Ras tumour overgrowth

Chiswili Chabu[1], Da-Ming Li[1] & Tian Xu[1,2]

Multiple signalling events interact in cancer cells. Oncogenic Ras cooperates with Egfr, which cannot be explained by the canonical signalling paradigm. In turn, Egfr cooperates with Hedgehog signalling. How oncogenic Ras elicits and integrates Egfr and Hedgehog signals to drive overgrowth remains unclear. Using a *Drosophila* tumour model, we show that Egfr cooperates with oncogenic Ras via Arf6, which functions as a novel regulator of Hh signalling. Oncogenic Ras induces the expression of Egfr ligands. Egfr then signals through Arf6, which regulates Hh transport to promote Hh signalling. Blocking any step of this signalling cascade inhibits Hh signalling and correspondingly suppresses the growth of both, fly and human cancer cells harbouring oncogenic Ras mutations. These findings highlight a non-canonical Egfr signalling mechanism, centered on Arf6 as a novel regulator of Hh signalling. This explains both, the puzzling requirement of Egfr in oncogenic Ras-mediated overgrowth and the cooperation between Egfr and Hedgehog.

[1] Department of Genetics, Howard Hughes Medical Institute, Yale University School of Medicine, Boyer Center for Molecular Medicine, 295 Congress Avenue, New Haven, Connecticut 06536, USA. [2] Children's Hospital & Institute of Developmental Biology and Molecular Medicine, Fudan University, Shanghai 20043, China. Correspondence and requests for materials should be addressed to T.X. (email: tian.xu@yale.edu).

Activating mutations of the *Ras* gene are highly prevalent in human cancers and give rise to some of the most aggressive tumours[1]. The molecular mechanisms governing oncogenic Ras-driven cancers are complex and involve interacting signalling pathways[1,2]. Paradoxically, oncogenic Ras cooperates with Egfr in cancers[3–6]. EGFR ligands bind to and activate Egfr, which recruits docking proteins via its cytoplasmic domain. Docking proteins (such as the downstream of receptor kinases or drk) activate the guanine exchange factor Son of sevenless (Sos), which converts Ras from an inactive (GDP-bound) to an active (GTP-bound) state and leads to the MAPK signalling cascade and activation of downstream target molecules[7,8]. It is surprising that the action of an activated downstream oncogenic component still requires its upstream receptor.

On the other hand, Egfr has been shown to cooperate with Hedgehog (Hh) signalling, another oncogenic pathway, to drive basal cell carcinoma and melanomas[9–12]. Hh signalling is initiated by the interaction of Hh protein with its receptor Patched. Endocytosis and intracellular transport of the receptor–ligand complex modulate Hh signalling levels[13]. How oncogenic Ras, Egfr and Hh signalling are integrated to concertedly drive tumour overgrowth remains unclear. Animal models expand our understanding of oncogenic Ras signalling.

Using a fly tumour model of oncogenic Ras we have identified Egfr as a positive regulator of oncogenic Ras-mediated overgrowth. Our characterization of Egfr's role in oncogenic Ras-mediated overgrowth led the finding that oncogenic Ras signalling stimulates the expression of the Egfr ligand spitz (spi) to recruit Egfr signalling and achieve tumour overgrowth. Egfr promotes tumour overgrowth independent of the canonical the Sos/Ras signalling, instead it acts via the ADP-Ribosylation Factor 6 (Arf6). Arf6 belongs to a family of highly conserved small Ras-related GTP-binding proteins and is largely known for its role in regulating endocytosis, vesicle transport and secretion[14–22]. We investigated a role for Arf6 in oncogenic Ras tumour overgrowth and found that Egfr promotes Arf6 to interact with Hh. This interaction allows Arf6 to control Hh cellular trafficking and promote Hh signalling. Consistent with this, blocking Egfr or Arf6 suppresses Hh signalling and inhibits the growth of either fly or human cancer cells harbouring oncogenic Ras. Altogether, our data delineate a non-canonical Egfr signalling mechanism in which Arf6 acts as a novel regulator of Hh signalling. This explains the puzzling requirement of Egfr in oncogenic Ras-mediated overgrowth and the oncogenic cooperation between Egfr and Hh signalling.

## Results

**$Egfr^-$ suppresses $Ras^{V12}$ tumours by inhibiting cell proliferation.** Mosaic expression of oncogenic *Ras* ($Ras^{V12}$) gives rise to hyperplastic tumours in *Drosophila* tissues[23,24]. These tumours can be GFP-labelled and a measure of the overgrowth phenotype can be readily obtained by examining the size and the fluorescence intensity of clones in dissected eye-antenna imaginal discs from third-instar animals[25,26] (Fig. 1a,b,f and g). We searched for mutations that suppress $Ras^{V12}$ tumour overgrowth[26] and identified a null *Egfr* mutation $Egfr^{Co}$ (ref. 27), hereafter referred to as *Egfr–*. We generated clones of cells harbouring the *Egfr–* mutation or expressing $Ras^{V12}$ in the presence of the *Egfr–* mutation and scored clones size. Consistent with EGFR's known role in controlling cell survival and growth[28], *Egfr–* mutant cells yielded small clones (Fig. 1a,c,f and h). We found that *Egfr–* suppressed $Ras^{V12}$ tumour overgrowth (Fig. 1a,b,d,f,g and i; quantified in Fig. 2j). A dominant-negative

version of Egfr (lacking its cytoplasmic tail)[29] produced a similar effect (Fig. 1e,j). Thus oncogenic Ras-mediated overgrowth requires the function of the upstream receptor Egfr.

We examined how Egfr cooperates with oncogenic Ras. Egfr could exert this tumour-promoting effect by regulating apoptosis and/or cell proliferation. We examined cell death in $Ras^{V12}$ or $Ras^{V12}$, *Egfr–* double mutant clones using terminal deoxynucleotidyl transferase dUTP nick end-labelling (TUNEL) assays. We did not detect ectopic cell death inside $Ras^{V12}$, *Egfr–* double mutant clones compared with $Ras^{V12}$ clones (Fig. 1k–n). Instead, *Egfr–* reduced the cell proliferation potential of $Ras^{V12}$ cells, as indicated by the decrease in the percentage of phospho-histone3 positive cells in $Ras^{V12}$, *Egfr–* double mutant clones compared with $Ras^{V12}$ clones (Fig. 1o,q versus p,r; quantified in 1s). Thus, Egfr fosters oncogenic Ras-mediated tumour overgrowth by promoting cell proliferation.

**$Ras^{V12}$ activates spitz to promote non-canonical Egfr signalling.** Oncogenic Ras could recruit Egfr's function by upregulating Egfr or its ligands. Oncogenic activation of the MAPK pathway has been shown to trigger the overexpression of Egfr and autocrine activation of Egfr by transforming growth factor alpha (TGF-α) family of Egfr ligands in cancer cells[4,6]. *Drosophila* has two TGF-α homologs, Gurken (Grk) and spitz (Spi)[30,31]. The expression and function of Grk are restricted to the germline in embryos, while Spi is broadly expressed and potently activates Egfr throughout all stages of development[30,32]. While we did not detect any change in Egfr protein levels in $Ras^{V12}$ clones (Fig. 2a), an antibody against the extra cellular domain of Spi showed elevated spi protein levels in and around $Ras^{V12}$ clones (Fig. 2b and b'). This argues that $Ras^{V12}$ cells upregulate Spi. Consistent with this, quantitative polymerase chain reaction analyses revealed that $Ras^{V12}$ transcriptionally stimulates *Spi* (Fig. 2k). Next, we assessed the functional significance of *Spi*'s upregulation on tumour overgrowth. We introduced a null *Spi* mutation (*spi–*) in $Ras^{V12}$ cells and scored clone sizes compared with $Ras^{V12}$ clones. Similar to *Egfr–*, the *spi–* mutation produced small clones and suppressed the overgrowth phenotype of $Ras^{V12}$ (Fig. 2c,d; quantified in j, and Supplementary Fig. 1d,g,j,m). In addition, we examined the effect of knocking down *Star*, which is required for Spi secretion and its growth control function (Supplementary Fig. 1a,c, and ref. 33). This also suppressed $Ras^{V12}$-mediated tumour overgrowth (Fig. 2c,e). Moreover, we took advantage of a conditional ligand-binding-defective *Egfr* allele, $Egfr^{tsla}$ (ref. 34). We examined the effect of this mutation on $Ras^{V12}$ tumour overgrowth at restrictive conditions and found that it also suppressed overgrowth (Fig. 2c,f, and Supplementary Fig. 1d,g,k,n). Collectively, these data argue that oncogenic Ras stimulates spi, which in turn triggers Egfr to increase tumour growth.

Egfr could contribute to oncogenic Ras-mediated overgrowth by augmenting canonical Egfr signalling levels via Ras[35]. Alternatively, Egfr signalling could exert its growth promoting effect via a previously uncharacterized mechanism. To distinguish between these two possibilities, we first tested whether knocking down Sos has an effect on $Ras^{V12}$ tumour overgrowth. While the expression of Sos-RNAi caused a small eye phenotype in adult animals, consistent with Egfr/Ras signalling inhibition (Supplementary Fig. 1a,b,d,g,l and o), it showed no effect on oncogenic $Ras^{V12}$ tumour overgrowth (Fig. 2c versus g). Similarly, direct inactivation of wild-type Ras using a null allele ($ras^{C40A}$) (ref. 36), produced small clones (Fig. 2h) but failed to suppress the $Ras^{V12}$ tumour overgrowth phenotype (Fig. 2c versus i). Taken together, the above data indicate that ligand-mediated Egfr function promotes oncogenic Ras tumour overgrowth independent of canonical Egfr signalling.

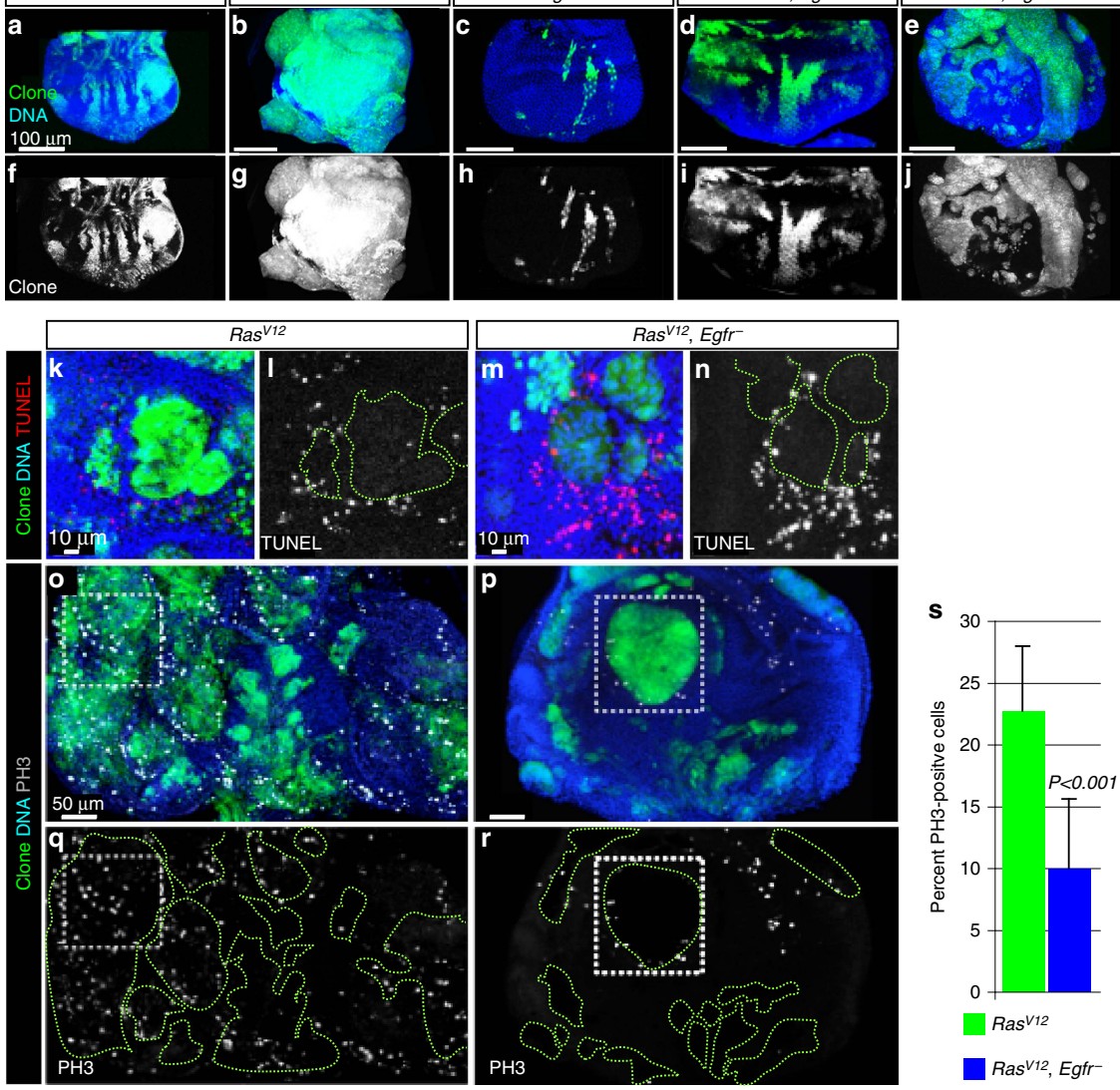

**Figure 1 | *Egfr*⁻ suppresses *Ras^V12* tumour overgrowth by inhibiting cell proliferation.** (a–j) Images of eye discs containing GFP-labelled wild-type or *Ras^V12* or *Egfr*⁻ single mutant or *Ras^V12*, *Egfr*⁻ or *Ras^V12*, *Egfr-DN* double mutant clones dissected from wondering third-instar animals raised at 25 °C. Images represent a projection of the top 10 μm for each genotype. *Ras^V12* clones (**b**) overgrow to form large contiguous tumours compared with control clones (**a**). The *Egfr*⁻ mutation yields small clones (**c**) and suppresses the growth of *Ras^V12* clones (**d**). Expression of a dominant version of Egfr (*Egfr-DN*) similarly suppresses *Ras^V12* tumour growth (**e**). Respective GFP (clones) channels are shown in the bottom panels (**f–j**). (**k–n**) Representative images showing terminal deoxynucleotidyl transferase (TdT) dUTP nick-end labelling (TUNEL) of *Ras^V12* or *Ras^V12*, *Egfr*⁻ double mutant clones to detect apoptotic cells. *Ras^V12*, *Egfr*⁻ double mutant clones (**m,n**) do not show ectopic cell death compared to *Ras^V12* clones (**k,l**). (**o–r**) Representative eye discs showing *Ras^V12* (**o**) or *Ras^V12*, *Egfr*⁻ double mutant (**p**) clones stained with anti-phosphohistone3 (PH3) antibodies to detect mitotic cells. Boxed areas are shown for comparison. Individual PH3 channels are shown in (**q,r**). (**s**) Quantitation of **o–r**. The number of PH3-positive over total number of cells was scored in multiple clones across several animals for each genotype. The *Egfr*⁻ mutation reduced the percentage of proliferative cells (Mean ± s.d.%, N, P: 9.97 ± 5.6%, N = 1,693 cells from 7 discs, P < 0.001 versus 27.7 ± 5.2%, N = 2,100 cells from 11 discs for *Ras^V12* and *Ras^V12*, *Egfr*⁻ double mutant cells, respectively). Error bars represent standard deviation (s.d.) from the mean for each genotype analysed. *P* is derived from *t*-test analyses and *N* denotes the sample size.

**Egfr promotes *Ras^V12*-mediated tumour overgrowth via ARF6.** In an effort to further elucidate how Egfr promotes oncogenic Ras-driven tumour overgrowth, we knocked down Egfr effectors (*Drk* and *Arf6*) by RNAi and asked whether this suppress oncogenic Ras-mediated tumour overgrowth, and identified Arf6 as a mediator of Ras tumour overgrowth. Arf6 belongs to a family of highly conserved small Ras-related GTP-binding proteins and mediates Egfr signalling in mammals and flies[15,37–39]. Arf6 knockdown in otherwise wild-type clones showed no detectable growth defects (Figs 1a and 3b) but potently suppressed *Ras^V12* tumour overgrowth (3a–c; quantified in 3e). In addition, Egfr directly recruits Arf6 GEFs to stimulate Arf6 (refs 37,38). We examined the effect of knocking the known Arf6 GEFs steppke and loner[38,40]. Knockdown of *loner*, but not *steppke*, suppressed *Ras^V12* tumour overgrowth (Fig. 3a,d–f, and Supplementary Fig. 1e,h), mimicking the effect of blocking the function of spi or Egfr or Arf6. In addition, we over-expressed Egfr in wing discs and asked whether knocking down Arf6 suppresses Egfr-mediated overgrowth effect and found that it does (Supplementary Fig. 2). Taken together, these data support the notion that Egfr acts through Arf6.

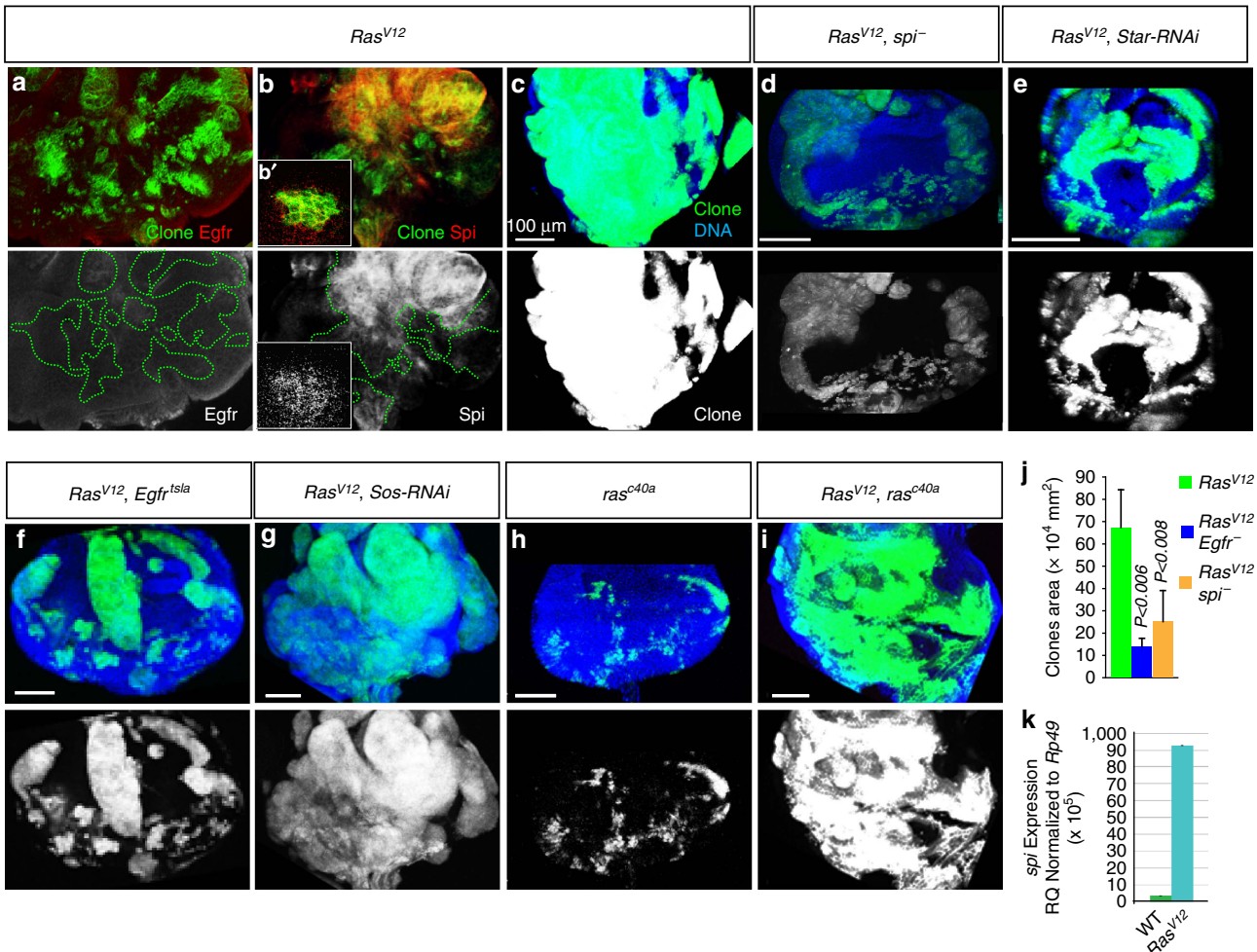

**Figure 2 | Oncogenic Ras upregulates the expression of the EGFR ligand *Spitz* to stimulate tumour overgrowth independent of canonical EGFR signalling.** (**a–b'**) Early third-instar eye discs showing $Ras^{V12}$ clones (green) stained with anti-Egfr or anti-spitz antibodies (red). Egfr protein levels remain unchanged in $Ras^{V12}$ clones (**a**), while spitz is specifically elevated in $Ras^{V12}$ clones and can be detected around some clones (**b,b'**). (**c–i**) Eye discs dissected from wondering third-instar animals showing clone growth for cells expressing $Ras^{V12}$ in otherwise wild-type background (**c**) or in cell carrying $spi^-$ (**d**) or $Egfr^{tlsa}$ (**f**) mutations or co-expressing *Star-RNAi* (**e**) or *Sos-RNAi* (**g**) or carrying the $ras^{c40a}$ (**i**) mutation. **h** is showing the growth of $ras^{c40a}$ clones in similarly aged animals. To the exception of $Ras^{V12}$, $Egfr^{tlsa}$ animals, which were raised at 29 °C, all animals were raised at 25 °C. $spi^-$ (**d**) *Star-RNAi* (**e**) and $Egfr^{tlsa}$ (**f**), but not *Sos-RNAi* (**g**), suppress the overgrowth of $Ras^{V12}$ clones (**b**). Similarly, $ras^{c40a}$ produced small clones (**h**) but fail to suppress the overgrowth of $Ras^{V12}$ tumour overgrowth (**i**). (**j**) Clone size quantitation. For this and all the subsequent clone size measurements, confocal stacks from similarly aged animal of the indicated genotypes and the image analysis software Imaris were used to measure clones areas. Mean ± s.d.%, N, P: $13.8 ± 3.5 × 10^4$, $N = 134$ clones, $P < 0.006$ ($Ras^{V12}$, $Egfr^-$ double mutant clones) or $24.9 ± 13.8 × 10^4$, $N = 91$ clones, $P < 0.008$ ($Ras^{V12}$, $spi^-$ double mutant clones) versus $67.1 ± 16.8 × 10^4$, $N = 20$ clones ($Ras^{V12}$ clones). (**k**) Reverse transcription PCR experiment measuring spi levels relative to expression levels of the housekeeping gene *Rp49*. Normalized fold differences of *spi* expression levels in eye discs bearing wild-type versus $Ras^{V12}$ clones. Experiments were performed in triplicates and standard deviations were derived from the coefficient variations of experimental and control samples. P is derived from *t*-test analyses and N denotes the sample size.

Given our fly tumour data, we tested whether Arf6 serves as a molecular link in the cooperation between oncogenic Ras and Egfr in human cancer cells. We knocked down Arf6 in various lung and colon cancer cell lines with oncogenic Ras mutations. We found that Arf6-RNAi specifically reduced Arf6 protein levels and suppressed the growth of these cancer cells in comparison to cancer cells without oncogenic Ras mutations or cells treated with RNAi control (Fig. 3g,h, and Supplementary Fig. 3a). We conclude that in both *Drosophila* and human cells, Arf6 has a critical role in the growth of Ras mutant tumours.

**Arf6 regulates Hh signalling to stimulate Ras tumour overgrowth.** We sought to define the mechanism by which Arf6 drives the overgrowth of $Ras^{V12}$ tumours. Hh signalling is a good candidate because it cooperates with Egfr to drive basal cell carcinoma and melanomas via an unknown mechanism[11,12]. Consistent with the possibility that Hh signalling could be involved in Egfr/Arf6-mediated Ras tumour overgrowth, we found that Hh was upregulated and co-localized with Arf6 in $Ras^{V12}$ clones (Supplementary Fig. 4). We assessed the status of Hh signalling by examining the expression levels of its transcriptional target *Cubitus interuptus* (*Ci* or *Gli* in vertebrates) in complementing biochemical and immunostaining experiments. Using antibody against active Ci, we found that Ci protein levels were elevated in lysates derived from dissected $Ras^{V12}$ discs compared with controls (Fig. 4a). Next, we directly stained wild-type or $Ras^{V12}$ mosaic tissues against *Ci*. In wild-type discs ci is suppressed posteriorly while Hh activates ci anteriorly (Fig. 4g: bottom and top brackets, respectively)[41]. In contrast,

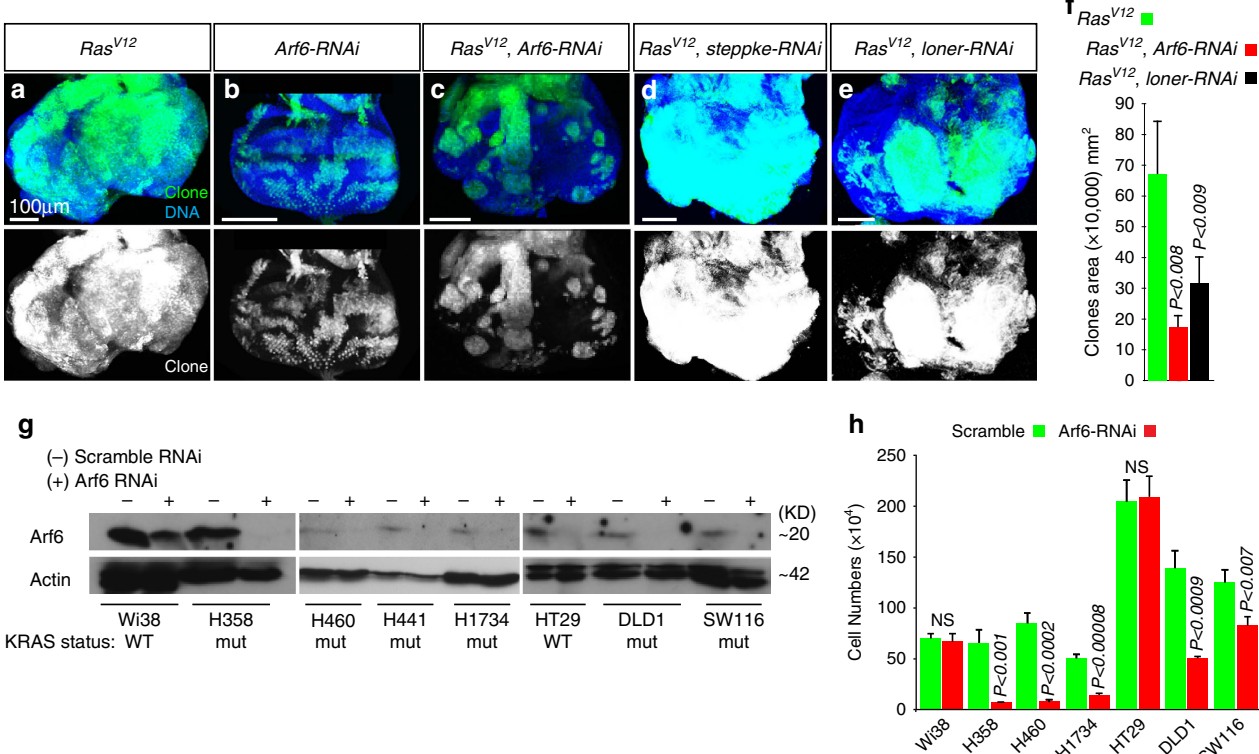

**Figure 3 | Arf6⁻ suppresses oncogenic Ras tumours in flies and human cancer cells.** (**a–e**) Representative images of wondering third-instar larval eye discs showing the growth of $Ras^{V12}$ (**a**) or $Arf6$-RNAi (**b**) clones or $Ras^{V12}$ clones co-expressing either $Arf6$-RNAi (**c**) or $steppke$-RNAi (**d**) or $loner$-RNAi (**e**). All animals were raised at 25 °C. $Arf6$-RNAi (**b**) causes no detectable growth defects but suppresses $Ras^{V12}$ tumour overgrowth (**c**). $loner$-RNAi (**e**), but not $steppke$-RNAi (**d**), blocks the overgrowth of $Ras^{V12}$ clones (**d**). (**f**) Quantitation of (**a–d**). Clones areas were $17.3 \pm 3.4 \times 10^4$, $N = 128$ clones, $P < 0.008$ ($Ras^{V12}$ clones expressing $Arf6$-RNAi) or $31.7 \pm 8.1 \times 10^4$, $N = 110$, $P < 0.009$ ($Ras^{V12}$ clones expressing $loner$-RNAi) versus $67.1 \pm 16.8 \times 10^4$, $N = 20$ clones ($Ras^{V12}$ clones). (**g**) Western blotting of various cancer cell lines treated with scramble or Arf6 RNAi and blotted with anti-Arf6 to detect Arf6 protein levels or anti-actin as a loading control. Cancer cells harbouring oncogenic Ras are denoted by $mut$ while cells with wild-type Ras are indicated by $WT$. ARF6 protein levels are considerably reduced in cells treated with Arf6 RNAi. (**h**) Quantitation of cell numbers of lung and cancer cell lines 2 days after transfection with either control (scramble) or Arf6 RNAi. Not significant (NS) denotes $P$-value $> 0.5$. $P$ is derived from $t$-test analyses and N denotes the sample size.

ci was upregulated in posterior $Ras^{V12}$ clones and this upregulation was more pronounced in larger clones (see discussion) (Fig. 4b,c, and corresponding bottom panels). In addition, expression of $Ras^{V12}$ in wing discs, showed ectopic ci activation (Supplementary Fig. 5). We conclude that oncogenic Ras activates Hh signalling.

The overexpression of ci in eye disc clones correlated with increased cell proliferation as determined by phospho-histone3 immunostaining (Fig. 4b and bottom panel). Knockdown of Arf6 suppressed Hh signalling in similar size clones and correspondingly reduced cell proliferation in $Ras^{V12}$ clones (Fig. 4b,c and t). Removing Spi or Egfr function in $Ras^{V12}$ clones ($Ras^{V12}$, $spi–$ or $Ras^{V12}$, $Egfr–$ double mutant cells) showed similar effects (Fig. 4b,c,e, and f), consistent with the notion that Egfr/Arf6 signalling stimulates Hh signalling in Ras tumours.

We have previously shown that tumours co-opt developmental mechanisms to promote overgrowth[25]. During development, dying cells upregulate Jun N-terminal kinase (JNK) signalling, which in turns instructs neighbouring cells to activate Janus kinase/signal transducers and activators of transcription (JAK-STAT) signalling and proliferate in order to maintain tissue homeostasis. $Ras^{V12}$ cells usurp this compensatory cell proliferation mechanism by forcing JNK activation in surrounding wild-type cells via elevated secretion of JNK signalling ligands, leading to tumour overgrowth[25]. We thus examined whether Arf6 regulates Hh signalling during

normal development. In the developing eye imaginal disc, anterior cells activate Hh signalling in response to Hh secreted from the posterior cells (Fig. 4h and ref. 41). Similarly, the $engrailed$ gene is specifically expressed in the posterior compartment (P) of the developing wing disc where it induces Hh expression but represses ci[42]. Posterior cells relay Hh to the anterior compartment (A) where Hh activates target genes including $ci$ and $patched$ ($ptc$) to control wing size and patterning[43,44]. We generated $Arf6$-RNAi clones in developing eye and wing discs, examined Hh signalling in the Hh responsive anterior cells, and found that $Arf6$-RNAi suppressed Hh signalling in both tissue types (Fig. 4g and bottom panel, bracket, and j; respectively). Conversely, we overexpressed $Arf6$ in wing discs using the dorsal driver $apterus$-GAL4 ($ap$-GAL4) and asked whether this upregulates Hh. Posteriorly produced Hh normally activates $ptc$ in a 8–10 cells diameter domain along the A/P boundary. Consistent with enhanced Hh signalling, $ap$-Gal4 > UAS-ARF6 discs showed elevated Ptc protein levels and an expansion of the Ptc domain specifically in the apterus-dorsal portion of the disc (Fig. 4k).

Next, we directly knocked down Arf6 along the A/P compartments boundary where high levels of Hh signalling regulate the patterning and growth of the wing. Arf6 knockdown interfered with the formation of the anterior cross vein and that of the longitudinal vein 3 (L3), a processes regulated by Hh signalling[45] (Fig. 4l,m, and arrowhead). Importantly, Arf6 knockdown

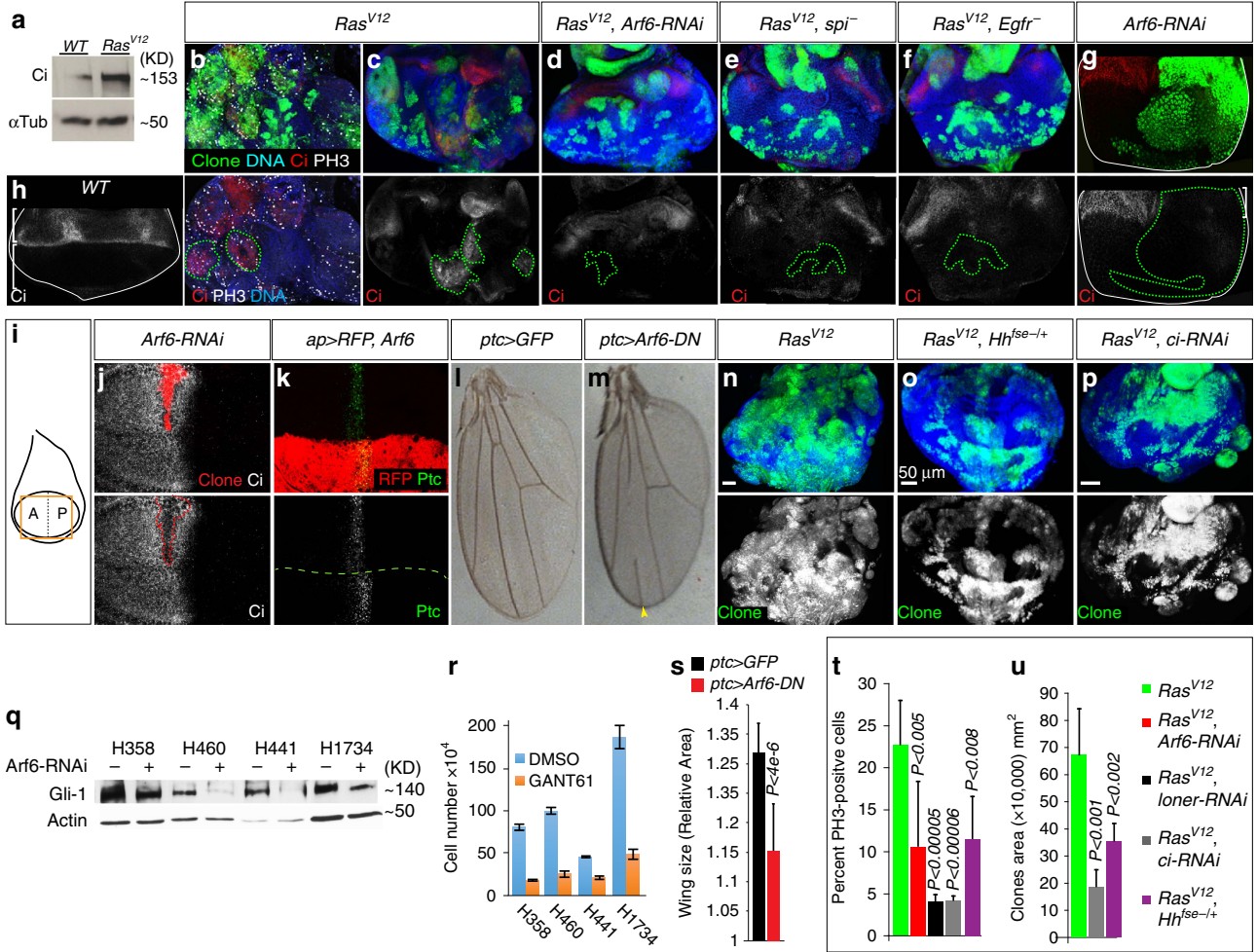

**Figure 4 | Arf6 regulates Hedgehog signalling to control tumour overgrowth. a**) Protein lysates derived from dissected wild-type discs or discs with *RasV12* clones were used in western-blotting experiments to detect active (full-length) Ci protein levels against the loading control (α-tubulin). (**b–f**) Early 3rd-instar eye discs showing *ey-FLP* MARCM clones of the indicated genotype stained with DAPI, anti-Ci, and/or anti-PH3. Bottom panels show PH3 and/or Ci separately. In this and all the remaining panels, dotted lines denotes clones boundary. (**g**) Early 3rd-instar eye disc showing *hs-FLP* MARCM *Arf6-RNAi* clones stained with anti-Ci. The bottom panel shows the Ci channel. **h**) Wild-type eye disc stained against Ci. The top bracket denotes the zone of high Hh signalling in the eye. (**i**) Schematic of a wing disc. The box illustrates the region of the disc shown in (**j** and **k**). **j**) Early 3rd-instar wing disc showing *hs-FLP Arf6-RNAi* MARCM clones (red) and stained against Ci. The Ci channel is shown in the bottom panel. (**k**) Expression of *Arf6-RNAi* in the dorsal domain (RFP) of the wing pouch in early 3rd-instar animals using *apterus-GAL4* and stained with anti-patched. Bottom panel shows patched channel alone. The dotted line delimits the dorsal-apterus domain in the wing pouch. (**l** and **m**) *Ptc-GAL4* expression of GFP or *Arf6-DN* (**n–p**) Similarly-aged eye discs showing clone growth for cells expressing *RasV12* alone (**n**) or with *Hhfse−/+* (**o**) or *ci-RNAi* (**p**). Clone channels are shown in bottom panels. (**q**) Western blotting of various lung cancer cell lines treated with scramble or Arf6 RNAi and blotted with anti-Gli1 or anti-actin. (**r**) Lung cancer cell lines treated either with DMSO or GANT61. (**s**) Quantitation of relative wing sizes from **o** and **p**. (**t**) Quantitation of the percentage of PH3-positive cells in clones of the indicated genotypes. (Mean ± SD%, *N*, *P*): 11.5 ± 5%, *N* = 1868, *P* < 0.008 (*RasV12*, *Hhfse−/+*) or 10.6 ± 7.7%, *N* = 2301, *P* < 0.005 (*RasV12*, *Arf6-RNAi*) or 4.1 ± 0.7%, *N* = 1764, *P* < 0.00005 (*RasV12*, *loner-RNAi*) or 4.15 ± 0.5 %, *N* = 6738, *P* < 0.00006 (*RasV12*, *ci-RNAi*) versus 27.7 ± 5.2%, *N* = 2100 for *RasV12*. **u**) Quantitation of (**n-p**). (Mean ± SD%, *N*, *P*): 35.3 ± 6.3x10$^4$ mm$^2$, *N* = 171 clones, *P* < 0.002 (*RasV12*, *Hhfse−/+*) or 18.5 ± 6.1x10$^4$ mm$^2$, *N* = 58 clones, *P* < 0.001 (*RasV12*, *ci-RNAi*). *P* is derived from *t*-test analyses and N denotes the sample size.

significantly reduced wing size compared with controls (*ptc-GAL4 > GFP*) (Fig. 4l,m,s, and Supplementary Fig. 6). Taken together, these data indicate that Arf6 regulates Hh signalling in Ras tumours and during development.

We subsequently tested whether Egfr/Arf6-triggered Hh signalling synergizes with oncogenic Ras to cause tumour overgrowth. We analysed the growth of *RasV12* clones induced in tissues heterozygotes for a hypomorphic allele of Hh, *Hhfse* (ref. 46), thus reducing Hh function by less than 50% compared with controls. We found that this considerably blocked cell proliferation and the overgrowth phenotype of *RasV12* clones (Fig. 4n,o; quantified in t and u). RNAi knockdown of ci showed

a similar effect (Fig. 4n,p; quantified in t and u). Egfr/Arf6-triggered Hh signalling thus synergizes with oncogenic Ras to cause tumour overgrowth.

Finally, we determined whether Arf6 similarly regulates Hh signalling in human cancer lines by examining changes in Gli1 expression levels following Arf6 RNAi knockdown. We focused on lung cancer cell lines because Arf6 knockdown potently inhibited growth in these cancer cells. Arf6 RNAi efficiently knocked down Arf6 protein levels (Fig. 3g) and reduced Gli1 levels in lung cancer cells (Fig. 4q and Supplementary Fig. 9). Direct inhibition of Hh signalling using GANT61, a specific Gli1 small molecule inhibitor[47,48], suppressed

growth similar to Arf6 knockdown (Fig. 4r). Taken together, we conclude that Arf6 regulates Hh signalling in both flies and human cancer cells to control growth.

**Arf6 controls Hh trafficking**. We investigated the relationship between EGFR, ARF6 and Hh using immunostaining and immunoprecipitation experiments. We previously observed that Arf6 co-localizes with Hh in $Ras^{V12}$ cells (Supplementary Fig. 4) and thus wondered whether Egfr modulates $Ras^{V12}$ tumour overgrowth by regulating an interaction between Arf6 and Hh. We performed Arf6 pull-down experiments in the presence or absence of Egfr and found that abrogating Egfr function in $Ras^{V12}$ cells diminishes Arf6's ability to pull-down Hh (Fig. 5a and Supplementary Fig. 7a), indicating that Egfr promotes the interaction of Arf6 with Hh. We then explored a mechanism for Arf6 regulation of Hh. Endocytosis and vesicle transport are known to control Hh signalling[13]. We hypothesized that Arf6 could promote the shuttling of Hh to signalling competent endosomes. In the absence of Arf6, Hh would be routed to the degradation pathway hence causing suppression of Hh signalling. The Hepatocyte growth factor-regulated tyrosine kinase substrate (Hrs) binds to and directs signalling molecules to the lysosomal degradation pathway to downregulate signalling[49]. We examined Hh localization to Hrs-positive vesicles and found that Arf6 knockdown caused Hh to predominantly localize to Hrs-positive vesicles compared with controls (Fig. 5b,e and f). Knocking down Egfr in $Ras^{V12}$ cells showed similar effects (Supplementary Fig. 7b–d). In addition, we directly used late endosomal markers (Arl8 and Rab7) to test whether Arf6 knockdown causes trafficking of Hh to lysosomal vesicles. We could not co-stain tissues with anti-Hh and anti-Arl8 (or anti-Rab7) antibodies because these antibodies were raised in the same species. Instead we used the *GMR-GAL4* driver to express GFP-labelled full-length or a version of Hh corresponding to its secreted and active form (Hh-GFP and Hh-N-GFP, respectively) in cells expressing either $Ras^{V12}$ alone or co-expressing *Arf6-DN* or *Arf6-RNAi,* and stained tissues against Arl8. We found that Hh-GFP preferentially localizes to clusters of Arl8-positive vesicles in $Ras^{V12}$, *Arf6-DN* tissues (Fig. 5c,c' and g). Similarly, we expressed Rab7-GFP in $Ras^{V12}$, *Arf6-DN* cells, stained against Hh, and found that Hh predominantly localizes to Rab7-GFP vesicles in these cells compared with controls ($Ras^{V12}$ cells) (Fig. 5d,h versus i,m). Moreover, expression of *Arf6-RNAi* in $Ras^{V12}$ cells resulted in the redistribution of Hh-N-GFP to Arl8 endosomes (Fig. 5j,n versus k,o). Furthermore, expression of *Arf6-RNAi* in otherwise wild-type cells similarly showed Hh localization to Rab7 endosomes (Supplementary Fig. 7e,f and h). Consistent with this, Patched (Hh receptor) localizes to Rab7 or Arl8 endosomes in *Arf6-RNAi* or *Arf6-DN* expressing cells (Fig. 5l,p, and Supplementary Fig. 6g–j). Together with the Hh-Hrs colocalization results above, these data indicate that Arf6 normally prevents the trafficking of Hh to the degradation pathway. Thus, Arf6 controls Hh trafficking in order to promote Hh signalling, which in turn cooperates with $Ras^{V12}$ to synergistically stimulate tumour overgrowth.

## Discussion

How oncogenic Ras signalling elicits Egfr and Hh functions and how these distinct signalling events are integrated to achieve oncogenic synergy in cancers is not well understood. Using a *Drosophila* tumour model we found that oncogenic Ras transcriptionally stimulates the TGF-α Egfr ligand spitz to recruit Egfr signalling and increase tumour growth. Egfr's role is likely tissue and/or context-dependent as inhibition of Egfr shows variable effects in different cancer types[4,5,50]. Interestingly,

Egfr inhibition in $Ras^{V12}$ appeared to cause a non-autonomous growth effect (Fig. 1a,d and e), suggesting that JNK signalling is involved[51]. However, blocking JNK signalling fails to rescue the growth of $Ras^{V12}$, $Egfr^-$ clones (Supplementary Fig. 8). Thus $Egfr^-$ suppresses $Ras^{V12}$ tumour overgrowth independent of JNK signalling. Instead, further analyses unexpectedly uncovered a non-canonical signalling modality for Egfr in $Ras^{V12}$ cells. Rather than signalling via the MAPK pathway, Egfr acts through Arf6. Knockdown of Arf6 in flies or human cancer cells suppresses the overgrowth of cell harbouring oncogenic Ras. This effect is due to a cell proliferation defect because, similar to Egfr knockdown, Arf6 knockdown showed no ectopic cell death in cancer lines with the highest growth suppression effect (Supplementary Fig. 2b).

We show that Arf6 acts via Hh signalling. Egfr promotes Arf6 to interact with Hh in order to stimulate Hh signalling. Arf6 stimulates Hh signalling by protecting Hh from lysosomal degradation. Consistent with this, Arf6 regulates the formation of carrier vesicles and interacts with lysosomes to control vesicle trafficking in epithelial cells[52,53]. We show that disrupting the interaction between Arf6 and Hh by either blocking Egfr or by directly depleting Arf6 causes Hh to accumulate in lysosomal vesicles. Interestingly, Arf6-depleted cells also show an increased number of Hrs-positive endosome overall, suggesting that Arf6 may impact other unknown signalling molecules in addition to Hh.

Hh signalling is activated in Ras tumour cells, and it is especially higher in larger clones. This could be because bigger clones produce more Spi and Hh in the clones' milieu allowing Arf6 to drive Hh signalling robustly in these cells. Importantly, blocking Egfr or Arf6 suppresses Hh signalling and correspondingly inhibits overgrowth in both flies and human cancer cells. Direct inhibition of Hh signalling suppresses the growth of fly and human cancer cells. Indeed we found that the overgrowth of Ras tumours is hypersensitive to Hh signalling dosage. Partial reduction of Hh protein is sufficient to effectively block tumour overgrowth, consistent with a synergetic cooperation. Thus Egfr/Arf6 has an important role in bridging oncogenic Ras and Hh signalling pathways.

Finally, we found that Arf6 regulates Hh signalling during normal development. ARF6 knockdown inhibited Hh signalling in developing imaginal discs and impinged on Hh signalling-regulated developmental processes.

Together our data highlight a non-canonical Egfr signalling mechanism centered on a novel role for Arf6 in Hh signalling regulation. We define a novel signalling mechanism that explains the perplexing requirement of Egfr in oncogenic Ras-mediated overgrowth and the oncogenic cooperation between Egfr and Hh signalling.

## Methods

**Experimental procedures**. *Fly lines*. Animals were aged at 25 °C on standard medium. The following fly lines were used in this study: (1) *y w; FRT82B*; (2) *y w; FRT82B, UAS-Ras^{V12}/TM6B*; (3) *Egfr^{Top−CA}* (T. Schupback, Princeton Univ.); (4) *UAS-DER(DN)* (A. Michelson, Harvard Univ.); (5) *UAS-spi-RNAi* (VDRC # 3922); (6) *cn1 Egf^{rsla} bw1/T(2;3)TSTL, CyO: TM6B, Tb1*(BDSC# 6501);(7) *FRT40, ras^{c40A}* (C. Berg, Univ. of Washington);*UAS-ARF6-RNAi* (VDRC # 24224, 100726; BDSC# 51417, 27261); (8) *UAS-loner-RNAi* (VDRC #106168 and BDSC#39060); (9) *UAS-ci-RNAi* (BDSC# 28984); (10) *UAS-Ras40A/CyO* (BDSC#114338); (11) *yw,ey-Flp1;act>y+>GAL4,UAS-GFP.S65T;FRT82B,tub-GAL80*; (12) *Hh^{fse}* (BDSC# 35562); (13)*yw,ey-Flp1;act>y+>GAL4,UAS-myrRFP;FRT82B,tub-GAL80*; (14) *yw,ey-Flp1; FRT42D,tub-GAL80; act>y+>GAL4,UAS-GFP*; (15) *UAS-ARF6-DN* (E. Chen, Johns Hopkins Univ.); (16)*yw,ey-Flp1;tub-GAL80,FRT40A;act>y+>GAL4,UAS-GFP.S65T*; (17) *Star-RNAi* (BDSC# 38914); (18) *Sos-RNAi* (BDSC#31174); (19) *Arf6-GFP* line is a previously generated and functionally validated GFP knock-in line[54,55] kindly provided by Y. Hong (University of Pittsburg). In brief, the *Arf6* gene is under endogenous control and a GFP is inserted in frame at the carboxyl terminus of Arf6 protein; (20) *UAS-steppke-RNAi* (BDSC#32374); (21) *UAS-Hh-GFP* and *UAS-Hh-N-GFP*, a gift from A. O'Reilly (Fox Chase Cancer Center, PA); (22) *yw, hsp70-Flp; act-y+ -GAL4, UAS-GFP*. See Supplementary Note 1 for genotype details of all images.

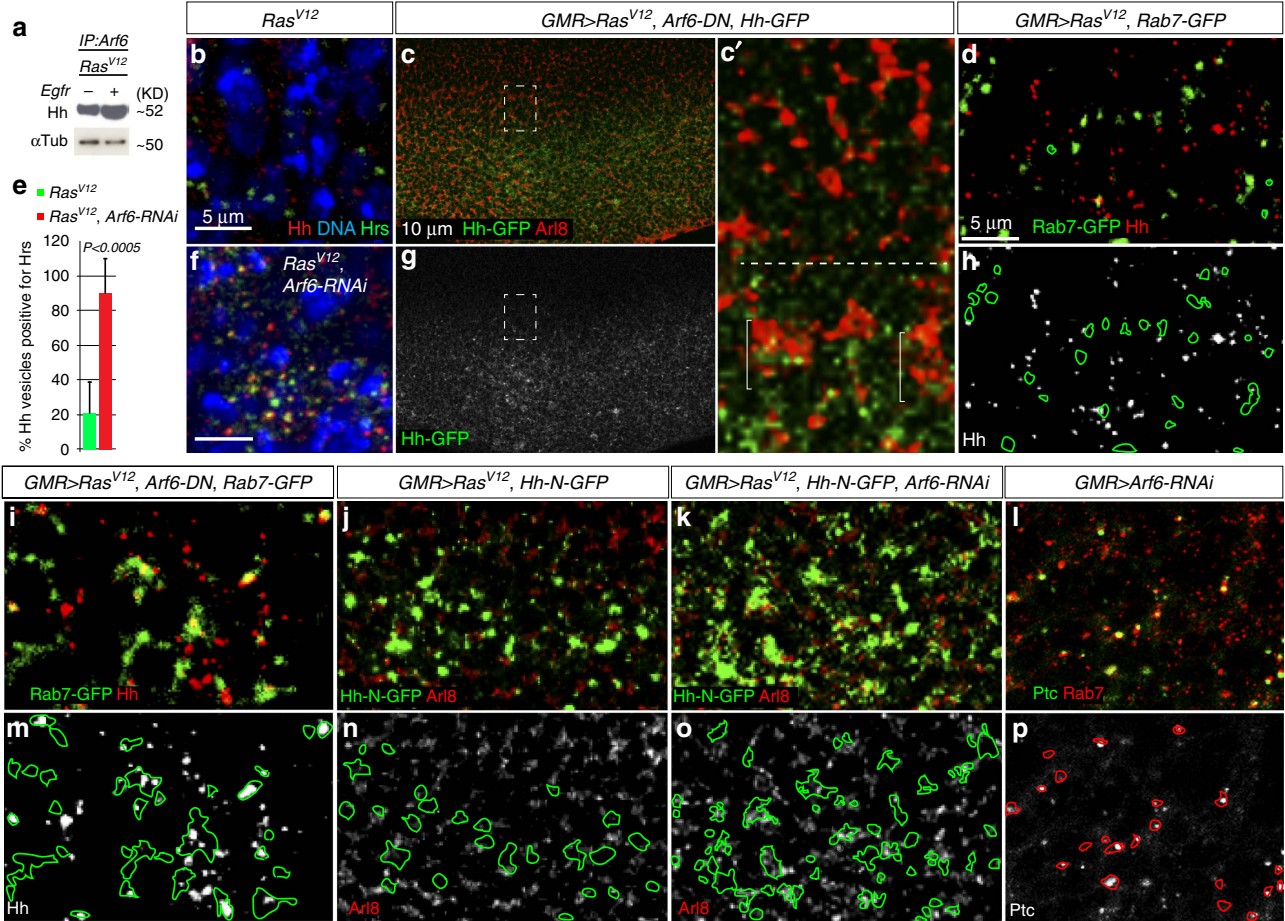

**Figure 5 | Arf6 controls Hedgehog cellular trafficking.** (**a**) Protein lysates derived from dissected wild-type discs or discs with $Ras^{V12}$ tumours were used in western-blotting experiments to detect full-length (active) Cubitus interuptus (Ci) protein levels against the loading control (α-tubulin). Hh co-precipitates with Arf6 but the $Egfr^-$ mutation interferes with Arf6's ability to interact with Hh. (**b,f**) $Ras^{V12}$ or $Ras^{V12}$, Arf6-RNAi co-expressing cells stained with anti-hedgehog, anti-Hrs antibodies and DAPI (4,6-diamidino-2-phenylindole). Distinct Hh (red) or Hrs (green)-positive vesicles are detected in $Ras^{V12}$ cells (**b**). In contrast, Hh predominantly localizes to Hrs-positive vesicles in $Ras^{V12}$, ARF6-RNAi co-expressing cells (**f**). (**c**) Eye disc co-expressing Hh-GFP, $Ras^{V12}$ and Arf6-DN under the control of GMR-GAL4 and stained against Arl8. The Hh-GFP channel alone is shown in **g**. Higher magnification image of posterior/anterior boundary (boxed area) is shown in **c′**. Hh-GFP localizes to clusters of Arl8-positive vesicles specifically in the posterior portion of the eye disc (brackets), but not in the anterior region of the eye (above the dotted line). (**d,h,l,m**) Images from eye disc co-expressing either $Ras^{V12}$ and Rab7-GFP (**d,h**) or $Ras^{V12}$, Arf6-DN, and Rab7-GFP (**i,m**) using GMR-GAL4 and stained with anti-Hh antibodies. Images of the Hh channel for **d,i** are shown in **h,m**, respectively. Fluorescent signal contours of Rab7-GFP (**m**) and Hh-N-GFP (**n,o**) are shown in **m–o**. (**e**) Quantitation of Hh punctae localizing to Hrs-positive vesicles. The total number of Hh punctae in $Ras^{V12}$ or $Ras^{V12}$, Arf6-RNAi co-expressing cells (N = 27 and 32, respectively) was scored for each genotype and the respective percentage of Hh localizing to Hrs-positive vesicles was determined. (**j,k**) Images from an eye disc co-expressing $Ras^{V12}$ and the secreted form of Hh (Hh-N-GFP) alone (**j**) or in the presence of Arf6-RNAi (**k**) and stained with anti-Arl8 antibodies. The respective Arl8 images are shown in the bottom panels **n,o**. (**l**) Image from an eye disc expressing Arf6-RNAi with GMR-GAL4 and stained against patched (ptc) and Rab7. The corresponding patched channel is shown alone in **p**. Rab7 fluorescent signal contours are shown in **p**. P is derived from t-test analyses and N denotes the sample size.

*Clones.* Clones were generated using standard MARCM techniques[56]. Detailed genotypes for all images are included in Supplementary Information. Briefly, all clones were induced with *ey-FLP* MARCM system, to the exception of Fig. 4g,j, which were generated with *hs-Flp*. For these particular experiments, 2 days larvae were heat-shocked for 1 h at 37 °C and raised for two days at 25 °C before processing. Images presented in Figs 2f, 4l,m, and Supplementary Figs 1k,n, 2, 5 and 6 were obtained from animal raised at 29 °C. For all remaining images, animals were raised at 25 °C. Clone size was obtained by determining the average clone area in discs dissected from randomly selected wondering third-instar control versus experimental larvae using the image analysis IMARIS. Properly aged GFP-positive mosaic animals were randomly and blindly selected under a dissecting microscope using white light to eliminate bias. Minimum sample sizes were determined using a 5% confidence level and 80% power. The minimum sample sizes were then used to estimate the number of animals/discs to be examined.

**Staining and imaging.** Eye-antenna discs were dissected, fixed and stained as described previously[25,26]. Tumour and adult eyes sizes analyses were carried out on a Leica MZ FLIII fluorescence stereomicroscope equipped with a camera.

Samples were examined by confocal microscopy with a Zeiss LSM510 Meta system and a Leica TCS SP8 confocal microscope. Images were analysed and processed with IMARIS (Bitplane, Switzerland) and Illustrator (Adobe) software, respectively. The following primary antibodies were used: guinea pig anti-Hrs (1:2,000, H. Bellen); Mouse anti-Spi (1:500; A.D. Vrailas-Mortimer); Rabbit anti-phosphohistone-H3 (1:1,000, Sigma); Rat anti-DER (1:1,000, B. Shilo, Weizmann Institute of Science/Israel); Mouse anti-EGFR (1:500, abcam); Rabbit anti-GFP (1:1,000, Abcam); Rabbit anti-Hh (1:500 Pre-absorbed, T. Kornberg, UCSF); Rat anti-Ci (detects the full length/active version of Ci, 1:200, Developmental Studies Hybridoma Bank, 2A1); Mouse anti-Patched (1:200, Developmental Studies Hybridoma Bank, Apa1). Secondary antibodies were from Invitrogen. TUNEL staining was performed using the *Apoptag Red* kit from Chemicon.

**Western blots and immunoprecipitation.** For immunoprecipitation experiments, imaginal discs from 50 third-instar animals carrying $Ras^{V12}$ single or $Egfr^{Top-CA}$, $RasV12$ or $DerDN(EGFR-DN)$, $Ras^{V12}$ double mutant clones were homogenized in lysis buffer (50 mM HEPES (pH 7.5), 150 mM KCl, 5 mM MgCl₂, 0.01% Triton-X)

supplemented with protease inhibitor tablets; Roche). Global protein concentrations was normalized across all genotypes and the lysate was pre-cleared with protein agarose-A beads for 1 h at 4 °C, incubated either with 2 µl of anti-Arf6 (M. Gonzalez-Gaitan) for 4 h at 4 °C. Lysates were then incubated with protein agarose A beads for 1 h at room temperature. For pull downs, beads were precipitated and washed three times in modified lysis buffer containing 0.5% TritonX-100. Samples were run on SDS–polyacrylamide gel electrophoresis, transferred on nitrocellulose membrane, and blotted with anti-Hh (T. Kornberg, UCSF) and anti-αTubulin (Sigma), loading control. For cancer cell lines, cells were lysed in lysis-loading buffer (0.5 M Tris (pH 6.8), 20% SDS, Glycerol, 10% β-mercaptoethanol and Bromophenol Blue), homogenized further by hydrodynamic shearing (15 times through a 25-gauge syringe, and centrifuged at 10,000 r.p.m. for 5 min. The supernatant was collected and run on SDS–polyacrylamide gel electrophoresis, transferred on nitrocellulose membrane, and blotted with anti-Gli1 (1/1,000; Abcam, cat# ab134906) or anti-Arf6 (1/1,000; Abcam, Cat# ab77581) or anti β-actin (1/1,000; Sigma, cat# A5441) antibodies. Full images of the blots are presented in Supplementary Fig. 9.

**Real-time polymerase chain reaction.** Total RNA from Eye-antenna imaginal discs containing wild-type or mutant clones was isolated using a Trizol RNA extraction method (Invitrogen). The SuperScriptIII First-Strand Synthesis System (Invitrogen) kit was used to synthesize complementary DNA. Real-time PCR was carried out on an Applied Biosystems machine using the SYBR green fast kit (Applied Biosystems) following the manufacturer's instructions. Relative gene expression was obtained from triplicate runs normalized to Rp49 as endogenous control. The following primers were used: Rp49, 5′-GGCCCAAG ATCGTGAAGAAG-3′ and 5′-ATTTGTGCGACAGCTTAGCATATC-3′; Spitz, 5′-CGCCCAAGAATGAAAGAGAG-3′ and 5′-AGGTATGCTGCTGGTG GAAC-3′

Clones of mutant cells in the eye-antennal discs were generated using standard MARCM system[56]. Immunostaining and Immunoprecipitation experiments were performed as previously described[57].

**ARF6 RNAi gene knockdown in human cancer cell lines.** In all, $2 \times 10^5$ cells were seeded onto 6-well-plates and treated with 2 nM of either ARF6 SiRNA (combination of three SiRNA of ARF6, Origene; Cat# SR300275) or universal control scrambled (Origene, SiRNA Cat# SR30004) and supplemented with 5 µl of DhamaFect 2 (GE- Dharmacon).

**Treatment of lung cancer cells with GANT61.** Cells were seeded onto 6-well-plates at a density of $2 \times 10^5$ and grown overnight before GANT61 (Cat# G9048-5MG, Sigma-Aldrich) treatment at the final concentration of $10 \, \mu m \, ml^{-1}$. Cell numbers were scored 96 h after GANT61 treatment.

**Data availability.** All data supporting the findings of this study are available within the article and its Supplementary Information files or from the corresponding author upon reasonable request.

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

## Acknowledgements

We thank T. Schupback (Princeton Univ. NJ), A. Michelson (Harvard Univ. MA), C. Berg (Univ. of Washington, WA), E. Chen (Johns Hopkins Univ. MD), Bloomington stock center, and the Vienna Drosophila RNAi Center for kindly providing reagents. This study was supported in part by grants from the National Basic Research Program of China (MOST2013CB945301) and from NIH/NCI to T.X. C.C. is funded by an NIH/NCI post-doctoral grant (1F32CA142118-01A1). T.X. is a Howard Hughes Medical Institute Investigator.

## Author contributions

C.C. and T.X. designed the research. C.C. performed experiments and analysed the data. D.-M.L. performed the mouse and human Arf6 RNAi knockdown experiments. C.C. and T.X. wrote the manuscript.

## Additional information

**Competing financial interests:** The authors declare no competing financial interests.

