## [Peer review file · Nature Communications]

Reviewers' comments:

Reviewer #1 (Remarks to the Author):

In this study, Chabu et al have investigated the mechanism by which EGFR cooperates with oncogenic Ras in tumorigenesis using the Drosophila model. The study reveals that Ras signalling upregulates the EGFR ligand, Spitz, which stimulates EGFR signalling via a non-canonical pathway via Arf6 to upregulate Hedgehog signalling. It is already known in mammalian cells that EGFR cooperates with oncogenic Ras, EGFR activates the Hedgehog pathway, and EGFR recruits Arf6-GEF to simulate Arf6 activity, however the study of Chabu et al., shows that this mechanism is conserved in Drosophila and more clearly delineates the mechanism to show that EGFR signalling leads to greater association of Arf6 with Hh and that inhibiting Arf6 results in an accumulation of Hh in Hrs-positive vesicles, which presumably then targets Hh for degradation in the lysosome. This aspect of the study seemed under-developed, and more mechanistic insight could have been obtained by some additional staining with endosomal markers. The manuscript is generally well-written, although would benefit from having subdivision of the Results section and headings that summarize the specific findings of each part. Furthermore, the Introduction needs to have a more comprehensive description of previous findings on Ras in tissue growth and tumorigenesis in Drosophila - at present the authors only cite their own paper from 2003, which is more to do with cooperation of RasV12 with polarity impairment. The Discussion section (needs to be labelled "Discussion") is just a summary of the results and needs to provide greater insight, for example 1) what other signalling pathways might Arf6 regulate by affecting vesicle trafficking that could be important in the RasV12 phenotype, 2) how does EGFR signalling activate Arf6-GEF - might this be through direct binding, and 3) what might be the developmental or physiological function of EGFR-Hh regulation? The quality of the data was high and mostly well quantified, although in the text further explanation is needed at several places - ie how were clones generated and were they in the eye or wing epithelium, and how was Arf6 identified in the screen alluded to in the results? Overall this is an interesting study that would be of great relevance to the signalling and cancer fields, however I believe that further tightening up of the data would improve the quality of the paper and its impact.

Specific comments:

- (1) Figures: The RasV12 MARCM clonal tissue seems to overtake the majority of the eye epithelium in some figures (1B, 2C) but not others (1O, 2A) - why is there variability - are larvae all staged at 5 days wandering 3rd instar and raised at the same temperature? This variability makes it important to quantify all interactions - eg the clonal area of RasV12 relative to with RasV12 with Star-RNAi, Sos-RNAi, and ras-c40e (Fig 2), to make all conclusions more robust.
- (2) Fig 1E - it seems that RasV12 EGFR-DN results in non-cell autonomous growth and indeed there seems to be Ph3-foci around the outside of the clones in 1P- is this the case, and why is this different from RasV12 EGFR mutant mosaic eye discs (Fig 1D)? [Fig 1O and 1P should have panels of Ph3 alone with the clones marked with dotted lines to make this easier to see.] RasV12 Arf6-RNAi and RasV12 loner-RNAi also seem to show non-cell autonomous overgrowth (Fig 3). This non-cell autonomous tissue growth effect, could be due to JNK upregulation and a secretory phenotype (Uhlirova et al., 2005, Nakamura et al., 2014)? This might be associated with the effects on Hrs accumulation seen in RasV12 Arf6-RNAi, and therefore controls should be done to show that JNK is not involved here.
- (3) Fig 2 - It would be good to show controls of spi knockdown, EGFR-*tsla*, Star-RNAi, Sos-RNAi and loner-RNAi clones alone with GFP and DAPI to see their effect alone on clonal growth (in a supp fig).
- (4) Fig 2 - how effective is the knockdown of Star - does it also result in an eye phenotype when knocked down in the whole eye, as does Sos?
- (5) Fig 4A legend should state that the samples were derived from ey-FLP MARCM mosaic eye-antennal tissue.
- (6) Fig 4F - The Arf6-RNAi mosaic eye shown is unusual with the clone taking up all of one half of the eye, and seems inconsistent with Arf6 being important for cell proliferation - was this a

common finding? Does Arf6-RNAi alone effect cell proliferation, or only in the context of RasV12?
(7) Fig 4N - How was the Arf6-RNAi clone generated in the wing disc? Only eyFLP is listed in the M&M.

(8) Fig 4P - did the Arf6-RNAi line also result in a wing vein ablation? The authors state that this is due to defective Hh signalling, but could it also be due to defective Ras signalling, given the role of Ras signalling in wing vein formation? Can this be rescued by expressing full-length Ci?

(9) Having shown Arf6 knockdown prevents the overgrowth of RasV12 expressing tissue it would be good to show whether elevated Arf6 signalling (expression of a Arf6-GTP locked form) cooperates with RasV12 to enhance tumorigenesis via increased Hh signalling?

(10) Fig 4W - the effect of the knockdown on the accumulation of Hrs vesicles and the colocalisation of Hh with Hrs was very interesting, but Hh might be expected to still be able to signal from the Multi-Vesicular Body (MVB), since Notch signalling still occurs in Escrt mutants, blocked at the MVB stage. This part of the study seemed incomplete, since other endosomal compartments, such as the early endosome, might accumulate in RasV12 tissue and be enriched with Hh, consistent with this compartment being important for promoting Hh signalling. Conversely, Arf6 knockdown might lead to greater lysosomal trafficking and degradation of Hh. Staining with early endosomal markers such as Rab5, Avl, late endosomal markers (Rab7) and lysosomal markers (Arl8) in RasV12 versus RasV12 Arf6-RNAi would provide more insight into this aspect of the study.

(11) Supp Fig 3 uses Arf6-GFP - this line needs to be described in the M&M.

(12) Introduction: References to the previous Drosophila literature on the effects of RasV12 expression in the eye should be listed - eg Karim & Rubin, 1998 and Halfar et al., 2001.

(13) Results section: P3 - It is not very precise to refer to RasV12 expressing clones as "tumors" - they are hyperplastic and have a competitive advantage, but differentiation still occurs and they are not invasive - so they should be termed "hyperplastic" or "benign tumors" so it is clear to the general reader.

(14) P3 - What does "unstable clones" mean? Would be better to describe them as "small clones" or "having reduced viability".

(15) P4 - The authors examine Spi, but Argos (negative regulator of EGFR signalling) has also been reported to be a target of Ras signalling - it would be interesting to see whether argos expression is also upregulated in RasV12 clones.

(16) P5 - A screen is mentioned as to how Arf6 was chosen for analysis, and the Materials and Method section is referred to for details of this, however I could not find any details there? This needs to be described.

(17) P5 - The effect of Arf6 knockdown on RasV12 mammalian cell proliferation was interesting, however it would add greater relevance if the connection of Arf6 to Hedgehog signalling could be made later in the paper (after the Hh section) in these mammalian cells and correlated with the effect on cell proliferation.

(18) P6 - The logic for investigating the Hh pathway because vesicle trafficking regulates Hedgehog signalling is not very convincing, since vesicle trafficking regulates many pathways including Notch. Best just to argue that they investigated the connection to Hh because of the mammalian evidence that Hh acts downstream of the EGFR.

(19) P6 - The Ci antibody needs to be described more fully in the results and M&Ms to state that it is to the full-length active version of Ci.

(20) P7 - More detail would be helpful in the statement - "we have previously shown that tumors co-opt developmental mechanisms to promote overgrowth"

(21) M&M - need to state at what temperature the experiments were conducted. RNAi lines result in greater knockdown at 29°C so presumably this temperature was used for the RNAi experiments?

(22) For all figures individual images of staining with the clones marked by dotted lines would help the reader more easily see the effects.

(23) Some extra labelling on figures would also help - ie Fig 4L, 4G - label "Ci", Fig Q-S bottom panels - label "GFP"

(24) For all Figures, state how the clones were induced for all genotypes in the figure legends.

Reviewer #2 (Remarks to the Author):

The manuscript by Chabu, Li and Xu addresses an enigma related to oncogenic Ras; despite being upstream of Ras, EGFR is required for Ras driven tumorigenesis. This observation points to involvement of a branch, downstream of EGFR, that is parallel to Ras and that contributes to activated-Ras induced phenotypes. The authors carry out a smart screen to identify this branch among the known EGFR effectors. They find that knockdown of Arf6, a Ras-related GTP-binding protein, can suppress oncogenic Ras driven overgrowth. They go on to show that Arf6 works via modulation of Hh signalling and this is the novel finding of the paper.

According to their model, activated Ras induces transcription of EGFR ligand Spitz, which leads to activation of EGFR and eventually Hh signalling via Arf6 which supposedly prevents/delays Hh degradation. Hh activation cooperates with and contributes to Ras-driven overgrowth. The model is neat as the circle is completed.

Overall, the data presented is of good quality and the findings are of interest to a wide readership. I would recommend publication in Nature Communications given the second half of the story, the connection between Arf6 and Hh signalling, is taken a bit further and the model is better supported.

Specific comments:

- The feedback regulation of Spi and its contribution to Ras driven overgrowth is neatly demonstrated. Is transcriptional regulation of Spitz by EGFR signalling shown for the first time here? Are examples of feedback regulation of its ligands via the EGFR pathway known in other systems? Please discuss.
- How important is Arf6 as an EGFR effector? Can it suppress activated-EGFR driven overgrowth?
- What is the role of Arf6 in vesicle trafficking?
- Can the authors comment more on the effect described in Suppl Fig. 3. Is this a regional effect? Which part of the disc are we looking at?
- Colocalization statements are difficult to make as only a small percentage of the signal actually overlap, but it might still be functionally relevant. Is there a difference between the RasV12 clones and wt tissue with respect to Hh and Arf6 colocalization?
- Genetic data showing that Hh signalling is important for the growth of RasV1 clones is very nice (Fig. 4R) while the claim that RasV12 leads to upregulation of Hh activity is less convincing. While I trust the authors conclusion is most likely correct, I would like to see better data points supporting this conclusion.
- In Figure 4, are panels H-L showing the same images as B-F? It is confusing! Would be nice to show the Ci channel alone for RasV12 experiment in B and H. This part of the paper is the least convincing part. Can the authors use other assays than immunostaining for Ci to measure Hh activity?
- Are these effects specific to the eye discs or seen in other discs as well?
Clones are by nature more difficult to interpret as they are randomly distributed. A simpler assay would be to induce RasV12 in a large area instead of clones, for example with Nub-G4 in the wing or a dorsal-eye-gal4 in the eye.
- It is nice that Arf6 RNAi treatment suppresses Ci levels in both eye and wing discs. Does RasV12 induce Ci in the wing?
- Can Arf6 overexpression stabilize Hh and increase Hh signalling?
- I disagree with the statement that "abrogating EGFR function in RasV12 cells suppresses ARF6's ability to pull-down Hh" on page 9 as still a good amount of Hh can be pulled down with ARF6.

- Does ARF6 knock-down or overexpression influence Hh localization to Hrs-positive vesicles in a wild-type background?
- The authors should use the same font size in all figures and panels throughout.

Reviewer #3 (Remarks to the Author):

Chabu et al, Nat Comm 2016

This manuscript presents evidence that oncogenic Ras acts in conjunction with EGFR signaling to drive cell proliferation in a *Drosophila* tumor model, and in a panel of human tumor cell lines. The data suggest that EGFR signaling is necessary for optimal proliferation even in the presence of oncogenic Ras, which is uncoupled from the activated receptor. The authors show that knockdown of either Arf6 or one of its guanine nucleotide exchange factors (Loner/Schizo) attenuates growth induced by oncogenic Ras, and that this may be due to altered trafficking of Hedgehog (Hh). The findings, particularly related to a potential role for Arf6 in Hh signaling, are novel and would be of interest to the cancer signaling community. However, there are several significant problems with the manuscript in its current form. These include:

1. Many of the images shown do not appear to match the corresponding quantitative data, and call into question how quantitation was performed. For example, in Fig. 2, panels A and C both represent control imaginal discs expressing oncogenic Ras. However, the staining pattern in A is very similar to the patterns in D, E and F, which supposedly represent attenuated growth caused by loss of Spitz (D), star (E) or ligand binding to EGFR (F). In addition, panel C is massively overexposed, while D appears to be underexposed relative to all the others. If A and C represent equivalent conditions, the error bars in the corresponding quantitation should be huge, and they are not. No methodology is provided describing how this quantitation was done. Since the entire study is based on quantitation of clone size, the data are impossible to interpret.

2. The authors use *Cubitus interruptus* (Ci) expression as a readout of Hh signaling. While Ci expression is indeed higher in some clones expressing oncogenic Ras, this is not true for all of them (Fig. 4B). How do the authors interpret this?

3. Arf6 is activated by a number of different GEFs. Although knockdown of Loner/Schizo does appear to weakly attenuate the growth of clones expressing oncogenic Ras (Fig. 3D), it is possible that it is also activated by other GEFs (e.g. Steppke). This should be examined more thoroughly, especially since they cite a paper describing a role for Steppke in EGFR signaling (Hahn et al, ref 27).

4. In the context of human cell lines, are all of the Ras-dependent lines also dependent on Hh signaling? Where does Hh come from under these conditions? If autocrine Hh signaling occurs in every case, this should be demonstrated.

5. In Fig. 4U, the authors use co-precipitation to suggest an interaction between Arf6 and Hh. However, there are two problems with this figure. First, total Hh should be shown in addition to tubulin. The difference in association +/- EGFR could simply be due to differences in overall expression level. Second, according to their Methods section, co-precipitation was performed from cell homogenates in the absence of detergent. Thus the two proteins may not interact at all, but are simply present in the same membranes.

6. The authors use colocalization of Hh with the endosomal protein Hrs as an indicator of entry into the degradative pathway. However, Hrs is present on early endosomes, and the image shown in Fig. 4W more likely represents accumulation of Hh in early, non-degradative endosomes in the absence of Arf6. How would this affect Hh signaling? Where is Patched (Hh receptor) under these conditions? Related to point #5, are Hh levels increased or decreased under these conditions? Where is the Hh under control conditions, if not associated with Hrs? And again, no description is provided of how quantitation of colocalization was performed. How many cells, how many discs were imaged to obtain the data shown in Fig. 4Z? Colocalization with bona fide late endosomal markers (mannose-6-phosphate receptors, cathepsins, LAMPs) would be necessary to make the authors' point here.

Response to Reviewers' Comments

MS #: NCOMMS-16-09682-T

MS TITLE: *EGFR/ARF6 Regulation of Hh Signaling Stimulates Oncogenic Ras Tumor Overgrowth*

Reviewer 1 Comments for the Author:

(1) Figures: The RasV12 MARCM clonal tissue seems to overtake the majority of the eye epithelium in some figures (1B, 2C) but not others (1O, 2A) - why is there variability - are larvae all staged at 5 days wandering 3rd instar and raised at the same temperature? This variability makes it important to quantify all interactions - eg the clonal area of RasV12 relative to with RasV12 with Star-RNAi, Sos-RNAi, and ras-c40e (Fig 2), to make all conclusions more robust.

Response: The reviewer is correct. The difference in clone size between Figures (1B, 2C) and Figures (1O, 2A) is because they are from different stages. Growth analyses for clones were conducted at wandering 3rd instar larval stage following standard practice (Figures 1B and 2C). Immuno-staining for assessing signaling, however, were carried out at early 3rd instar stage when clones are in rapidly growing phase and can be compared with surrounding wild-type tissue (Figures 1O and 2A). Mutant and wild-type control animals were raised at the same condition. Clone size from animals at the same stage is quantified. We apologize for not making clear that growth analysis and immune-staining were from different time points and have now specified this in all relevant figure legends.

(2) Fig 1E - it seems that RasV12 EGFR-DN results in non-cell autonomous growth and indeed there seems to be Ph3-foci around the outside of the clones in 1P- is this the case, and why is this different from RasV12 EGFR mutant mosaic eye discs (Fig 1D)? [Fig 1O and 1P should have panels of Ph3 alone with the clones marked with dotted lines to make this easier to see.] RasV12, Arf6-RNAi and RasV12, loner-RNAi also seem to show non-cell autonomous overgrowth (Fig 3). This non-cell autonomous tissue growth effect, could be due to JNK upregulation and a secretory phenotype (Uhlirova et al., 2005, Nakamura et al., 2014)? This might be associated with the effects on Hrs accumulation

seen in RasV12 Arf6-RNAi, and therefore controls should be done to show that JNK is not involved here.

Response: The apparent difference between tissues with *Ras^{V12}*, *EGFR-DN* clones (Figure 1D) and tissues with *Ras^{V12}*, *Egfr⁻* mutant clones (Figure 1E) are because the previous two images had different scale bars in the old Figure 1. The tissues with either type of clones behave similarly and both show non-autonomous overgrowth. We have now used images with the same scale bars to avoid the confusion (new Figure 1D, 1E).

Following the reviewer's suggestion, we have now shown PH3 channel alone with the clones marked by dotted lines (Figure 1Q, 1R).

We agree that determining whether JNK signaling mediates *EGFR-DN* effects on *Ras^{V12}* growth is important. We tested this by directly blocking JNK signaling in *Ras^{V12}*, *EGFR-DN* clones and asked whether this would rescue growth and found that it did not. This indicates that *EGFR-DN* suppresses *Ras^{V12}*-mediated clone overgrowth independent of JNK signaling. We have included this data in Supplemental Figure and discussed this in the discussion (Page13, lines 271-275; Supplementary Fig. 8).

(3) Fig 2 - It would be good to show controls of spi knockdown, EGFR-tsla, Star-RNAi, Sos-RNAi and loner-RNAi clones alone with GFP and DAPI to see their effect alone on clonal growth (in a supp fig).

Response: We now have included these data in Supplementary Figure 1d-o.

(4) Fig 2 - how effective is the knockdown of Star - does it also result in an eye phenotype when knocked down in the whole eye, as does Sos?

Response: The *Star* knockdown results in a smaller eye, similar to *Sos* knockdown. This data is now included in Supplementary Figure 1a-c.

(5) Fig 4A legend should state that the samples were derived from ey-FLP MARCM mosaic eye-antennal tissue.

Response: We have added this information in Figure 4 legend and in the Materials/Methods section (page 27, lines 666-667 and page 16, lines 333-336; respectively).

(6) Fig 4F - The Arf6-RNAi mosaic eye shown is unusual with the clone taking up all of one half of the eye, and seems inconsistent with Arf6 being important for cell proliferation- was this a common finding? Does Arf6-RNAi alone effect cell proliferation, or only in the context of RasV12?

Response: While *ARF6-RNAi* invariably suppresses the growth of *Ras^{V12}* clones, it shows no obvious effect on cell proliferation on its own as judged by mutant clone size (Figure 3b).

(7) Fig 4N - How was the Arf6-RNAi clone generated in the wing disc? Only eyFLP is listed in the M&M.

Response: *ARF6-RNAi; FRT82B* animals were crossed to the *yw, ey-Flp1; act>y+>GAL4,UAS-GFP.S65T; FRT82B, tub-GAL80*. We are now including the complete genotype for all the images in Supplementary information.

(8) Fig 4P - did the Arf6-RNAi line also result in a wing vein ablation?

The authors state that this is due to defective Hh signalling, but could it also be due to defective Ras signalling, given the role of Ras signalling in wing vein formation? Can this be rescued by expressing full-length Ci?

Response: *ARF6-RNAi* does have a weak wing vein ablation phenotype (Supplementary Fig. 6). To test whether Hh signaling can rescue *ARF6-DN* vein phenotype, we co-expressed *ARF6-DN* and full-length Ci (*Ptc>ARF6-DN, Ci^{ACT}*) as the reviewer suggested. However, these animals do not eclose, making the analysis impossible.

(9) Having shown Arf6 knockdown prevents the overgrowth of RasV12 expressing tissue it would be good to show whether elevated Arf6 signalling (expression of a Arf6-GTP locked form) cooperates with RasV12 to enhance tumorigenesis via increased Hh signalling?

Response: Ectopic expression of constitutive active ARF6 (ARF6-CA) did not further enhance the growth of *Ras*^{V12} clones, suggesting that ARF6 growth enhancing effect is maximized in *Ras*^{V12}.

(10) Fig 4W - the effect of the knockdown on the accumulation of Hrs vesicles and the colocalisation of Hh with Hrs was very interesting, but Hh might be expected to still be able to signal from the Multi-Vesicular Body (MVB), since Notch signalling still occurs in *Escrt* mutants, blocked at the MVB stage. This part of the study seemed incomplete, since other endosomal compartments, such as the early endosome, might accumulate in *Ras*^{V12} tissue and be enriched with Hh, consistent with this compartment being important for promoting Hh signalling. Conversely, *Arf6* knockdown might lead to greater lysosomal trafficking and degradation of Hh. Staining with early endosomal markers such as Rab5, Avl, late endosomal markers (Rab7) and lysosomal markers (Arl8) in *Ras*^{V12} versus *Ras*^{V12} *Arf6*-RNAi would provide more insight into this aspect of the study.

Response: Following the reviewer's suggestion, we used late endosomal markers (Arl8 and Rab7) to test whether ARF6 knockdown causes trafficking of Hh to lysosomal vesicles, as suggested by Hh protein localization to Hrs-positive vesicles in ARF6 depleted *Ras*^{V12} cells. We could not co-stain tissues with anti-Hh and anti-Arl8 (or anti-Rab7) antibodies because these antibodies were raised in the same species. Instead we used the *GMR-GAL4* driver to express GFP-labeled full-length or a version of Hh corresponding to its secreted and active form (Hh-GFP and Hh-N-GFP, respectively) in cells expressing either *Ras*^{V12} alone or co-expressing *ARF6-DN* or *ARF6-RNAi*, and stained tissues against Arl8. We found that Hh-GFP preferentially localizes to clusters of Arl8-positive vesicles in *Ras*^{V12}, *ARF6-DN* tissues (Figure 5c, 5c', and 5g). Similarly, we expressed Rab7-GFP in *Ras*^{V12}, *ARF6-DN* cells, stained against Hh, and found that Hh predominantly localizes to Rab7-GFP vesicles in these cells compared to controls (*Ras*^{V12} cells) (Figure 5d, 5h vs. 5i, 5m). Moreover, expression of *ARF6-RNAi* in *Ras*^{V12} cells resulted in the redistribution of Hh-N-GFP to Arl8 endosomes (Figure 5j, 5n vs. 5k, 5o). Furthermore, expression of *ARF6-RNAi* in otherwise wild-type cells similarly showed Hh localization to Rab7 endosomes (Supplementary Fig. 7 e, f, and h). Consistent with this,

patched (Hh receptor) localizes to Rab7 or Arl8 endosomes in *ARF6-RNAi* or *ARF6-DN* expressing cells (Fig. 5l, 5p, and Supplementary Fig. 6 g-j). Together with our previous results with Hrs, our data strongly argue that ARF6 normally prevents the trafficking of Hh to the degradation pathway. These new data have now been included in the manuscript (pages 12 and 13, lines 241-258).

(11) Supp Fig 3 uses Arf6-GFP - this line needs to be described in the M&M.

Response: The ARF6-GFP line is a previously generated and functionally validated GFP knock-in line ^{1,2}. In brief, the *ARF6* gene is under endogenous control and a GFP is inserted in frame at the carboxyl terminus of ARF6 protein. This information has now been added to the Materials and Methods section (page 16, lines 325-328).

(12) Introduction: References to the previous *Drosophila* literature on the effects of RasV12 expression in the eye should be listed - eg Karim & Rubin, 1998 and Halfar et al., 2001.

Response: Both references are included in on page 4, lines 58-59.

(13) Results section: P3 - It is not very precise to refer to RasV12 expressing clones as "tumors" - they are hyperplastic and have a competitive advantage, but differentiation still occurs and they are not invasive - so they should be termed "hyperplastic" or "benign tumors" so it is clear to the general reader.

Response: We agree and have now clearly stated that *Ras^{V12}* clones give rise to "hyperplastic tumors" in the introduction before simply referring them as tumors. This information is on page 4, lines 58-59.

(14) P3 - What does "unstable clones" mean? Would be better to describe them as "small clones" or "having reduced viability".

Response: We agree and are now describing *Egfr* clones as small clones (page 4, line 67).

(15) P4 - The authors examine Spi, but Argos (negative regulator of EGFR signaling) has

also been reported to be a target of Ras signaling - it would be interesting to see whether argos expression is also up-regulated in RasV12 clones.

Response: We stained tissues containing *Ras*^{V12} clones against Argos and found that Argos protein level was not elevated in *Ras*^{V12} clones compared to surrounding wild-type tissues (see below).

Argos expression in *Ras*^{V12} clones

A-C) Images showing eye disc containing *Ras*^{V12} clones (green) stained with DAPI and anti-Argos antibodies (C). Individual clone channel is shown in (B).

(16) P5 - A screen is mentioned as to how Arf6 was chosen for analysis, and the Materials and Method section is referred to for details of this, however I could not find any details there? This needs to be described.

Response: We did not describe or mention a screen in the manuscript. ARF6 is a result of a candidate approach.

(17) P5 - The effect of Arf6 knockdown on RasV12 mammalian cell proliferation was interesting, however it would add greater relevance if the connection of Arf6 to Hedgehog signalling could be made later in the paper (after the Hh section) in these mammalian cells and correlated with the effect on cell proliferation.

Response: We examined changes in Gli1 (Hh signaling transcriptional target) protein levels following ARF6 RNAi knockdown and found that ARF6 RNAi reduced GLI1 levels in multiple lung cancer cell lines (Fig. 4q). We then directly blocked Gli1 activity in these cells with GANT61, a specific Gli1 small molecule inhibitor^{3,4} and found that it suppresses growth (Fig. 4r), similar to ARF6 knockdown. Thus, similar to the effects of ARF6 knockdown in flies, ARF6 knockdown blocks Hh signaling in mammalian cells

and this effect correlates with growth inhibition. These exciting new data have been added in Figure 4q, r, and in the text (page 11; lines 213-219).

(18) P6 - The logic for investigating the Hh pathway because vesicle trafficking regulates Hedgehog signalling is not very convincing, since vesicle trafficking regulates many pathways including Notch. Best just to argue that they investigated the connection to Hh because of the mammalian evidence that Hh acts downstream of the EGFR.

Response: We added the additional reference from mammalian studies showing there is a synergy between Hh and EGFR signaling (page 8; lines 151-153).

(19) P6 - The Ci antibody needs to be described more fully in the results and M&Ms to state that it is to the full-length active version of Ci.

Response: We have included this information in material and methods (page 17, lines 354-355).

(20) P7 - More detail would be helpful in the statement - "we have previously shown that tumors co-opt developmental mechanisms to promote overgrowth"

Response: We have added more details on this. During development, dying cells upregulate JNK signaling which in turns instructs neighboring cells to activate JAK-STAT signaling and proliferate in order to maintain tissue homeostasis. *Ras*^{V12} cells usurp this compensatory cell proliferation mechanism by forcing JNK activation in surrounding wild-type cells via elevated secretion of JNK signaling ligands, leading to tumor overgrowth. This information is on page 9, lines 175-181.

(21) M&M - need to state at what temperature the experiments were conducted. RNAi lines result in greater knockdown at 29°C so presumably this temperature was used for the RNAi experiments?

Response: All RNAi experiments were performed at 25°C, to the exception of the wing size assays and experiments involving *Egfr*^{Δs}, which were carried out at 29°C. We apologize for not making this clear. The information has been added to the Materials and Methods section (page 16, lines 338-340).

(22) For all figures individual images of staining with the clones marked by dotted lines would help the reader more easily see the effects.

Response: We have included individual channels with dotted lines representing clones boundary in all our staining images.

(23) Some extra labelling on figures would also help - ie Fig 4L, 4G - label "Ci", Fig Q-S bottom panels - label "GFP"

Response: We have included labels.

(24) For all Figures, state how the clones were induced for all genotypes in the figure legends.

Response: We now include detailed genotypes for all images in Supplementary Information.

Reviewer #2

1) How important is Arf6 as an EGFR effector? Can it suppress activated-EGFR driven overgrowth?

Response: We thank the reviewer for suggesting this experiment. We ectopically expressed EGFR in the Ptc stripe in the wing discs and found that it causes overgrowth compared to controls, as expected (Supplementary Fig. 6a and 6b). Expression of *ARF6-RNAi* significantly suppressed the overgrowth phenotype (Supplementary Fig. 6c), indicating that ARF6 is an important effector of EGFR. This information has been added on pages 7 and 8, lines 136-138.

2) What is the role of Arf6 in vesicle trafficking?

Response: ARF6 regulates the formation of carrier vesicles to positively or negatively control vesicle trafficking in epithelial cells endosomes⁵⁻⁷. Our data show that ARF6 prevents Hh from entering the degradation pathway. This is supported by our additional data showing that Hh predominantly localizes to degradation pathway endosomes

(Rab7/Arl8-positive) in ARF6 knockdown cells. This information has been added to the discussion section on page 14; lines 283-288.

3) Can the authors comment more on the effect described in Suppl Fig. 3. Is this a regional effect? Which part of the disc are we looking at?

Response: We found that clones of *Ras^{V12}* cells show elevated Hh protein levels compared to surrounding wild-type cells in eye discs irrespective of the position of the clones. We have replaced the images with lower magnification images to help readers better assess the effect of *Ras^{V12}* on Hh expression (Supplementary Fig. 3).

4) Colocalization statements are difficult to make as only a small percentage of the signal actually overlap, but it might still be functionally relevant. Is there a difference between the *Ras^{V12}* clones and wt tissue with respect to Hh and Arf6 colocalization?

Response: ARF6 and Hh colocalize in *Ras^{V12}* cells as well as in wild-type cells (see panel (A) in the figure below). To further illustrate this, we examined ARF6-Hh colocalization in wild-type developing wing discs. We found that ARF6-GFP consistently co-localizes with Hh (Panels B-E, bracket in the figure below).

ARF6-GFP/Hh colocalization in eye and wing discs.

A) Image showing tissue expressing ARF6-GFP and containing clone of *Ras^{V12}* cells delimited with dotted line and stained with Hh and DAPI. ARF6-GFP is a GFP knock-in line expressing full-length C-terminal GFP tagged ARF6 under the endogenous promoter.

B) Schematic of a wing disc. The green box illustrates the region of the disc shown in (B-D).

B-D) x/z cross sections view of the A/P boundary of wild-type discs expressing ARF6-GFP under endogenous control and co-stained with Hh (red) and DAPI (blue).

5) Genetic data showing that Hh signalling is important for the growth of RasV12 clones is very nice (Fig. 4R) while the claim that RasV12 leads to upregulation of Hh activity is less convincing. While I trust the authors conclusion is most likely correct, I would like to see better data points supporting this conclusion.

Response: In addition to the initial image and the Western blot (Fig. 4a) showing elevated Ci levels in clones and in tissue lysates, respectively, we have now included an additional image to show elevated Hh signaling in Figure 4c. In a separate experiment suggested below by the reviewer, ectopic expression of *Ras^{V12}* using *nub-GAL4* showed ectopic Ci activation in wing discs (Supplementary Fig. 5), providing further evidence that oncogenic Ras stimulates Hh signaling.

6) In Figure 4, are panels H-L showing the same images as B-F? It is confusing! Would be nice to show the ci channel alone for rasV12 experiment in B and H. This part of the paper is the least convincing part. Can the authors use other assays than immunostaining for Ci to measure Hh activity?

Response: We apologize for the confusion. We have added an image showing Ci upregulation in Ras clones (Fig. 4c). Image of the individual Ci channel with clone boundary is shown in the bottom panel, as the reviewer suggested. Detailed description of the panels has also been added to the figure legends to avoid confusion (page 27, lines 666-670).

Ci immunostaining is routinely used and represents the standard assay for assessing Hh signaling status. In addition to Ci immunostaining, we have also used a complementing biochemical approach and found that Ci protein levels were elevated in *Ras^{V12}* cells compared to control from Western blot experiments (Fig. 4a).

7) Are these effects specific to the eye discs or seen in other discs as well? Clones are by nature more difficult to interpret as they are randomly distributed. A simpler assay would be to induce RasV12 in a large area instead of clones, for example with Nub-G4 in the wing or a dorsal-eye-gal4 in the eye. It is nice that Arf6 RNAi

treatment suppresses Ci levels in both eye and wing discs. Does RasV12 induce Ci in the wing?

Response: We followed the reviewer's suggestion and expressed *Ras^{V12}* in wing discs using *nub-Gal4* and stained against active Ci to assess Hh signaling. Ci activation is restricted to the anterior compartment in wild-type discs. In contrast, expression of *Ras^{V12}* resulted in ectopic Ci activation in the posterior compartment, providing additional evidence that oncogenic Ras activates Hh signaling. This information has been added in Supplementary Figure 5 and in the text (Page 9, lines 165-167).

9) Can Arf6 overexpression stabilize Hh and increase Hh signalling?

Response: We thank the reviewer for suggesting this experiment. To test whether ARF6 stabilizes Hh signaling, we overexpressed ARF6 in wing discs using the dorsal driver *apGAL4*, and asked whether this up-regulates Hh signaling by staining tissues against Ci. Posteriorly produced Hh normally activates Ptc in a 8-10 cells diameter domain along the A/P boundary. Consistent with enhanced Hh signaling, *apGal4>UAS-ARF6* discs showed elevated Ptc protein levels and an expansion of the Ptc domain specifically in the dorsal portion of the disc. This data has been added in Figure 4k and in the text (page 10, lines 191-196) as it nicely complements ARF6 loss of function phenotype.

10) I disagree with the statement that "abrogating EGFR function in RasV12 cells suppresses ARF6's ability to pull-down Hh" on page 9 as still a good amount of Hh can be pulled down with ARF6.

Response: We have substituted "suppresses" with "diminishes".

11) Does ARF6 knock-down or overexpression influence Hh localization to Hrs-positive vesicles in a wild-type background?

Response: We extended our analysis of Hh localization using additional endosomal markers (Rab7 and Arl8) and found that ARF6 knockdown in otherwise wild-type cells causes Hh to localize to Rab7/Arl8-positive endosomes. This new data is included in Figure 5l, p, and in Supplementary Figure 7 e, f, and h.

12) The authors should use the same font size in all figures and panels throughout.

Response: We apologize for this and have done this.

Reviewer #3 (Remarks to the Author):

1. Many of the images shown do not appear to match the corresponding quantitative data, and call into question how quantitation was performed. For example, in Fig. 2, panels A and C both represent control imaginal discs expressing oncogenic Ras. However, the staining pattern in A is very similar to the patterns in D, E and F, which supposedly represent attenuated growth caused by loss of Spitz (D), star (E) or ligand binding to EGFR (F).

Response: We apologize for the confusion. The difference in clone size between Figures 2a and 2c is because they were from different developmental stages. Growth analyses for clones were conducted at wandering 3rd instar larval stage following standard practice (Fig. 2c-i). Immuno-staining for assessing signaling, however, were carried out at early 3rd instar stage when clones are in rapid growing phase and can be compared with surrounding wild-type tissue (Fig. 2a). All clone size quantification data are derived from analyzing discs of wandering 3rd instar animals. We apologize for not making this clear and have now labeled this in the figure legends (page 25, lines 617 and 621).

In addition, panel C is massively overexposed, while D appears to be underexposed relative to all the others. If A and C represent equivalent conditions, the error bars in the corresponding quantitation should be huge, and they are not. No methodology is provided describing how this quantitation was done. Since the entire study is based on quantitation of clone size, the data are impossible to interpret.

Response: The image in Figure 2c is not overexposed. Both Figures 2c and 2d were acquired under identical imaging conditions. The difference in fluorescence intensity comes from the fact that *Ras^{V12}* clones (Fig. 2c) are significantly overgrown compared to *Ras^{V12}, spi* double mutant clones (Fig. 2d). On the other hand, Figure 2a and 2c do not represent equivalent conditions. As mentioned above, growth analyses for clones were

conducted at wandering 3rd instar larval stage following standard practice (Fig. 2c). Immuno-staining for assessing signaling, however, were carried out at early 3rd instar stage when clones are in rapid growing phase and can be compared with surrounding wild-type tissue (Fig. 2a). We apologize for not making clear and have now labeled this in the figure legends (page 25, lines 617 and 621).

Clone size was obtained by determining the average clone area in control versus experimental animals using the image analysis IMARIS ©. This information is now included in Materials and Methods (page 16, lines 340-341). All the relevant statistical information is included in the figure legend.

2. The authors use *Cubitus interruptus* (Ci) expression as a read-out of Hh signaling. While Ci expression is indeed higher in some clones expressing oncogenic Ras, this is not true for all of them (Fig. 4B). How do the authors interpret this?

Reviewer: The reviewer is correct, not all clones show high Ci levels. In fact, Hh signaling was particularly high in bigger, more proliferative clones. This could be because bigger clones would produce more Spi and Hh in the clones' milieu allowing ARF6 to drive a more robust Hh signaling in these cells. We have included this in the discussion (page 14, lines 291-293).

3. Arf6 is activated by a number of different GEFs. Although knockdown of Loner/Schizo does appear to weakly attenuate the growth of clones expressing oncogenic Ras (Fig. 3D), it is possible that it is also activated by other GEFs (e.g. Steppke). This should be examined more thoroughly, especially since they cite a paper describing a role for Steppke in EGFR signaling (Hahn et al, ref 27).

Response: Following the reviewer's suggestion, we used a previously validated RNAi line⁸⁻¹⁰ and knocked-down Steppke in *Ras*^{V12} cells and found that it had no discernible effect on growth. This information has been included in Figure 3d and in the text (page 7, lines 131-135).

4. In the context of human cell lines, are all of the Ras-dependent lines also dependent on Hh signaling? Where does Hh come from under these conditions? If autocrine Hh signaling occurs in every case, this should be demonstrated.

Response: We examined whether Hh signaling is required in human cancer cell lines and found that it is. We directly blocked Gli1 activity in multiple lung cancer cell lines with GANT61, a specific Gli1 small molecule inhibitor^{3,4} and found that it suppresses the growth of all the tested cell lines. Hh comes from the cancer cells, as Hh expression is upregulated in lung cancer cells¹¹. This exciting new data has been added in Figures 3g, 4q and in the text (page 11; lines 213-219).

5. In Fig. 4U, the authors use co-precipitation to suggest an interaction between Arf6 and Hh. However, there are two problems with this figure. First, total Hh should be shown in addition to tubulin. The difference in association +/- EGFR could simply be due to differences in overall expression level. Second, according to their Methods section, co-precipitation was performed from cell homogenates in the absence of detergent.

Thus the two proteins may not interact at all, but are simply present in the same membranes.

Response: We apologize for inadvertently omitting to indicate that the lysis buffer contained detergent (.01% TritonX-100). However, we repeated the experiment with detergent-containing lysis buffer again and blotted for total Hh protein in the pre-precipitation lysates and in the post-precipitation eluates. Although the total protein concentrations in the lysates was much lower this time because of the laborious dissections it requires to obtain highly concentrated lysates from larval discs and the time constraints on resubmission, we observed that EGFR knockdown diminished ARF6's ability to co-precipitate with Hh. This experiment is included in Supplementary Figure 4a.

6. The authors use colocalization of Hh with the endosomal protein Hrs as an indicator of entry into the degradative pathway. However, Hrs is present on early endosomes, and the image shown in Fig. 4W more likely represents accumulation of Hh in early, non-

degradative endosomes in the absence of Arf6. How would this affect Hh signaling? Where is Patched (Hh receptor) under these conditions? Related to point #5, are Hh levels increased or decreased under these conditions? Where is the Hh under control conditions, if not associated with Hrs? And again, no description is provided of how quantitation of colocalization was performed. How many cells, how many discs were imaged to obtain the data shown in Fig. 4Z? Colocalization with bona fide late endosomal markers (mannose-6-phosphate receptors, cathepsins, LAMPs) would be necessary to make the authors' point here.

Response: Following the reviewer's suggestion, we used late endosomal markers (Arl8 and Rab7) to test whether ARF6 knockdown causes trafficking of Hh to the degradation pathway, as suggested by Hh protein localization to Hrs vesicles in *ARF6* depleted *Ras^{V12}* cells. There is no available working *Drosophila* cathepsins or mannose-6-phosphate receptors antibodies for immuno-staining. We used anti-Rab7 and anti-Arl8 antibodies instead, as suggested by the other reviewers. We could not co-stain tissues with anti-Hh and anti-Arl8 (or anti-Rab7) antibodies because these antibodies were raised in the same species. Instead we used the *GMR-GALA* driver to express GFP-labeled full-length or a version of Hh corresponding to its secreted and active form (Hh-GFP and Hh-N-GFP, respectively) in cells expressing either *Ras^{V12}* alone or co-expressing *ARF6-DN* or *ARF6-RNAi*, and stained tissues against Arl8. We found that Hh-GFP preferentially localizes to clusters of Arl8-positive vesicles in *Ras^{V12}*, *ARF6-DN* tissues (Figure 5c, 5c', and 5g). Similarly, we expressed Rab7-GFP in *Ras^{V12}*, *ARF6-DN* cells, stained against Hh, and found that Hh predominantly localizes to Rab7-GFP vesicles in these cells compared to controls (*Ras^{V12}* cells) (Figure 5d, 5h vs. 5i, 5m). Moreover, expression of *ARF6-RNAi* in *Ras^{V12}* cells resulted in the redistribution of Hh-N-GFP to Arl8 endosomes (Figure 5j, 5n vs. 5k, 5o). Furthermore, expression of *ARF6-RNAi* in otherwise wild-type cells similarly showed Hh localization to Rab7 endosomes (Supplementary Fig. 7 e, f, and h). Consistent with this, patched (Hh receptor) localizes to Rab7 or Arl8 endosomes in *ARF6-RNAi* or *ARF6-DN* expressing cells (Fig. 5l, 5p, and Supplementary Fig. 6 g-j). Together with our previous results with Hrs, our data strongly argue that ARF6 normally prevents the trafficking of Hh to the degradation pathway. These new data have now been included in the manuscript (pages 12 and 13, lines 241-258).

Ras^{V12}, *ARF6-RNAi* or *Ras^{V12}*, *Egfr⁻* and *Ras^{V12}* clones show similar Hh expression.

For the Hh/Hrs colocalization study, we analyzed 27 wild-type or 32 *Ras^{V12}*, *ARF6-RNAi* cells across 3 and 5 discs, respectively. The quantification in Figure 4z (Fig. 5e in this new version) is obtained from scoring the total number of Hh punctae and deriving the percentage of Hh punctae positive for Hrs in cells of each genotype. This information has been added to the figure legend (page 30, lines 733-736).

Reference

- 1 Huang, J., Zhou, W., Dong, W. & Hong, Y. Targeted engineering of the Drosophila genome. *Fly (Austin)* **3**, 274-277 (2009).
- 2 Huang, J., Zhou, W., Dong, W., Watson, A. M. & Hong, Y. From the Cover: Directed, efficient, and versatile modifications of the Drosophila genome by genomic engineering. *Proceedings of the National Academy of Sciences of the United States of America* **106**, 8284-8289, doi:10.1073/pnas.0900641106 (2009).
- 3 Lauth, M., Bergstrom, A., Shimokawa, T. & Toftgard, R. Inhibition of GLI-mediated transcription and tumor cell growth by small-molecule antagonists. *Proceedings of the National Academy of Sciences of the United States of America* **104**, 8455-8460, doi:10.1073/pnas.0609699104 (2007).
- 4 Agyeman, A., Jha, B. K., Mazumdar, T. & Houghton, J. A. Mode and specificity of binding of the small molecule GANT61 to GLI determines inhibition of GLI-DNA binding. *Oncotarget* **5**, 4492-4503, doi:10.18632/oncotarget.2046 (2014).
- 5 Peters, P. J. *et al.* Characterization of coated vesicles that participate in endocytic recycling. *Traffic* **2**, 885-895 (2001).
- 6 Maranda, B. *et al.* Intra-endosomal pH-sensitive recruitment of the Arf-nucleotide exchange factor ARNO and Arf6 from cytoplasm to proximal tubule endosomes. *The Journal of biological chemistry* **276**, 18540-18550, doi:10.1074/jbc.M011577200 (2001).
- 7 Hurtado-Lorenzo, A. *et al.* V-ATPase interacts with ARNO and Arf6 in early endosomes and regulates the protein degradative pathway. *Nat Cell Biol* **8**, 124-136, doi:10.1038/ncb1348 (2006).
- 8 Lee, D. M., Wilk, R., Hu, J., Krause, H. M. & Harris, T. J. Germ Cell Segregation from the Drosophila Soma Is Controlled by an Inhibitory Threshold Set by the Arf-GEF Steppke. *Genetics* **200**, 863-872, doi:10.1534/genetics.115.176867 (2015).

- 9 Liu, J., Lee, D. M., Yu, C. G., Angers, S. & Harris, T. J. Stepping stone: a cytohesin adaptor for membrane cytoskeleton restraint in the syncytial *Drosophila* embryo. *Mol Biol Cell* **26**, 711-725, doi:10.1091/mbc.E14-11-1554 (2015).
- 10 Lee, D. M. & Harris, T. J. An Arf-GEF regulates antagonism between endocytosis and the cytoskeleton for *Drosophila* blastoderm development. *Curr Biol* **23**, 2110-2120, doi:10.1016/j.cub.2013.08.058 (2013).
- 11 Hong, Z. *et al.* Activation of hedgehog signaling pathway in human non-small cell lung cancers. *Pathol Oncol Res* **20**, 917-922, doi:10.1007/s12253-014-9774-x (2014).

REVIEWERS' COMMENTS:

Reviewer #1 (Remarks to the Author):

In this revised submission of their Manuscript, Xu and colleagues have adequately addressed the reviewers comments and added new data, which substantially improves the paper. I would support acceptance of this important manuscript.

Some minor points that would improve this version of the manuscript are listed below:

(1) hs-FLP is not described in the fly stocks in the M&M. Should be listed as hsp70-FLP in M&M and then abbreviated as (hs-FLP)

(2) p5 - Although a screen was not done, it would be helpful to reader if there was a some description of what other EGFR effector candidate genes were analysed? Also I dont understand why is it necessary to refer to the M&M for the Arf RNAi lines here?

(3) In relation to Figure 2 - What is not clear is how many samples were analysed per genotype? This should be indicated in the fig leg.

(4) The authors should check for typographic errors (eg EFGR in abstract - change to EGFR) and that all genetic elements are in italics and correct nomenclature as in Flybase.

Reviewer #2 (Remarks to the Author):

I think, the authors made a great effort to address all the reviewers comments and I support the publication of this work.

Few aspects could still be improved (or taken into account in preparation of future manuscripts):

- Do not crop reviewer's comments. This prompted me to find my full report to make sure that every point was addressed and I noticed that my first point was not addressed. (was not a major point)

- I agree with reviewer 3 that in many panels (1B, 2C, 3D) the GFP signal is greatly overexposed generating ugly images. I do not think taking all the images at the same settings justify this as these images are not suitable for quantitative analysis.

Reviewer #3 (Remarks to the Author):

The authors have done a good job of addressing my earlier concerns, and a substantial amount of new data has been added to reinforce their conclusions. No additional work is necessary.

Response to Reviewers' Comments

MS #: NCOMMS-16-09682-T

MS TITLE: *EGFR/ARF6 Regulation of Hh Signaling Stimulates Oncogenic Ras Tumor Overgrowth*

Reviewer #1

(1) hs-FLP is not described in the fly stocks in the M&M. Should be listed as hsp70-FLP in M&M and then abbreviated as (hs-FLP)

Response: We listed the hs-FLP line as “*yw, hsp70-Flp; act-y⁺-GAL4 UAS-GFP*” (Xu and Rubin, 1993) in Materials/Methods and have abbreviated it as (*hs-FLP*). (Page 16, lines 388 and 395; page 28, lines 766; and page 29, line 775).

(2) p5 - Although a screen was not done, it would be helpful to reader if there was a some description of what other EGFR effector candidate genes were analysed? Also I dont understand why is it necessary to refer to the M&M for the Arf RNAi lines here?

Response: We agree with the reviewer. We have now included the genes that were analyzed and no longer refer to the Material/Methods for the *Arf6-RNAi* lines. See Page 7, lines 175-178.

(3) In relation to Figure 2 - What is not clear is how many samples were analyzed per genotype? This should be indicated in the fig leg.

Response: Although we indicated the number of cells examined for each genotype, we agree with the reviewer that we should have also included the number of samples analyzed. This information has been added to the figure legend. (Page 25, lines 706-707).

(4) The authors should check for typographic errors (eg EFGR in abstract - change to EGFR) and that all genetic elements are in italics and correct nomenclature as in Flybase.

Answer: We thank the reviewer for catching that. We have corrected typographic errors and have properly noted genetic elements consistent with Flybase nomenclature.

Reviewer #2:

- Do not crop reviewer's comments. This prompted me to find my full report to make sure that every point was addressed and I noticed that my first point was not addressed. (was not a major point)

Response: Sometimes there are multiple questions in one paragraph. We broke the paragraph into multiple parts to try to answer each question. We apologize for inadvertently omitting a question.

-I agree with reviewer 3 that in many panels (1B, 2C, 3D) the GFP signal is greatly overexposed generating ugly images. I do not think taking all the images at the same settings justify this as these images are not suitable for quantitative analysis.

Response: The panels in question (1B, 2C, and 3D) were not singularly overexposed. All growth analyses images within figures were acquired with the same setting. The reason why Ras clones are particularly bright is because they grow far more than clones of the other genotypes.

While Ras clones occupy the majority of the eye discs in projection images (Figure legend: pages 25, lines 688-689), the clones are however discreet and their size can be quantified in images of confocal slices.

Reference

Xu, T., Rubin, G.M., 1993. Analysis of genetic mosaics in developing and adult *Drosophila* tissues. *Development* 117, 1223-1237.